# *Dimorphilus gyrociliatus* (Annelida: Dinophiliformia) Dwarf Male Nervous System Represents a Common Pattern for Lophotrochozoa

**DOI:** 10.3390/biology11111674

**Published:** 2022-11-17

**Authors:** Elizaveta Fofanova

**Affiliations:** Department of Comparative and Developmental Physiology, Koltzov Institute of Developmental Biology RAS, 119334 Moscow, Russia; lizchenbio@mail.ru or fofanova@idbras.ru

**Keywords:** dwarfism, serotonin, nervous system, annelida, apical organ, trochophore, miniaturization, lophotrochozoa

## Abstract

**Simple Summary:**

The dwarf male of *D. gyrociliatus* is the smallest known annelid, at only 28 μm × 40 μm in size. Data on its morphology are limited. This study provides a description of the external ciliation and the morphology of the serotonin-like immunoreactive nervous system of the dwarf male. Serotonin (5-HT) is known to play a key role in controlling reproduction, ciliary locomotion, and sensing in Annelida. Immunochemical labeling with anti-acetylated tubulin, anti-5-HT antibodies, and confocal microscopy were used to visualize the ciliary structures and individual neurons. The male has three ciliary fields. The nervous system of the male includes frontal ganglia, five commissures, two ventral and two dorsal bundles, and penial nerves. A total of 57 neurons were found, of which only five were 5-HT-like immunoreactive. There may be homology between the male and female ciliary structures. The dwarf male may also possess a compact apical organ consisting of two sensory neurons and one 5-HT-like immunoreactive cell.

**Abstract:**

Dinophiliformia is a newly revealed clade within Annelida that is a sister group to Pleistoannelida. *Dimorphilus gyrociliatus* is a representative of this clade that has fascinated scientists with its high degree of sexual dimorphism. Both males and females are small, worm-like creatures that have specific ciliary structures: anterior ventral, posterior ventral, and dorsal ciliary fields in males, and prototroch, metatroch, and ventral ciliary fields in females. There are data on the morphology and development of the nervous system in Oweniidae, Sipunculida, Pleistoannelida, and even Dinophiliformia. However, data on the neuromorphology and development of *D. gyrociliatus* dwarf males are limited. Here, we present data on the distribution of cilia, sensory neurons, and the 5-HT-like immunoreactive system in 3D reconstructions and cross-sections. Immunochemical labeling with anti-acetylated tubulin and anti-5-HT antibodies and confocal microscopy were used to visualize the ciliary structures and neurons. The male has three ciliary fields: anterior ventral, posterior ventral, and dorsal. These include frontal ganglia, five commissures, two ventral and two dorsal bundles, and penial nerves. A total of fifty-seven neurons and only five 5-HT-like immunoreactive cells were described. Although the sensory neurons were not 5-HT-like immunoreactive, they had 5-HT innervation, which may indicate the role of 5-HT in perception. There may be homology between male and female ciliary structures. The dwarf male of *D. gyrociliatus* may have a reduced apical organ consisting of two sensory neurons and a 5-HT-like immunoreactive cell.

## 1. Introduction

Dinophiliformia is a newly revealed clade within the Annelida group (Figure 1) [1,2]. This group includes miniature species that were difficult to place in the phylogenetic tree. Dinophiliformia is related to Errantia and Sedentaria (Pleistoannelida) [1,2].

The representatives of this group show different morphological patterns, even within one species. *Dimorphilus gyrociliatus* (formerly *Dinophilus gyrociliatus*) is one of these species. It is famous for its strong sexual dimorphism, which is already evident during embryogenesis. Adult females lay cocoons with larger and smaller eggs; the smaller eggs develop into males and the larger eggs develop into females. Embryonic development takes about a week in females and less in males. After hatching, dwarf males immediately begin mating. Both males and females are small, worm-like creatures that have specific ciliary structures: anterior ventral, posterior ventral, and dorsal ciliary fields in males, and prototroch, metatroch, and ventral ciliary fields in females. Adult females are about 700-1000 microns long and have a simplified morphology with six segments, without parapodia or chaetae. Dwarf males are highly diminutive, at only 40 μm long, and lack a mouth and digestive system while possessing a sensory system and specialized glandomuscular copulatory organ [3,4].

While data on the morphology and development of the nervous system are available for Oweniidae [5,6], Sipunculida [7,8,9], Pleistoannelida (Errantia [10] and Sedentaria [11]), and even Dinophiliformia (*D. vorticoides* both sexes and *D. gyrociliatus* females [12]), data on *D. gyrociliatus* dwarf males are limited.

Some aspects of the morphology of mature dwarf males of *D. gyrociliatus* have already been described using confocal, scanning and transmission electron microscopy [3,4,13]. However, there are limited data on external ciliation and the organization of the nervous system in terms of transmitter content and 3D reconstruction. The most detailed data to date were presented by Westheide and Windoffer using SEM and TEM [3,4]. Their results showed that the mature dwarf male has three prominent ciliary fields: the anterior ventral, the posterior ventral, and the anterior dorsal ciliary field (Figure 2). Each ciliary field is composed of multiciliated cells. The authors analyzed the number of cells in these fields and invented a special nomenclature and numbering system [4]. The anterior ciliary field consists of seven pairs of multiciliated cells arranged symmetrically on the left and right sides of the ventral surface (*E1–E7*); the posterior ventral ciliary field consists of two pairs of multiciliated cells (*E8, E9*); and the dorsal ciliary field includes two pairs of multiciliated cells (*E10, E11*) (Figure 2) [4].

As for internal organization, the dwarf male has a frontal gland, a pair of protonephridia, prominent testis, and a specialized copulatory system that includes gonoducts, stylet gland cells, adhesive glands, and the copulatory organ itself [3].

The authors also gave an accurate and detailed description of the nervous system, which includes about 68 neuronal cell bodies. Most of these neurons were classified as sensory based on their position and morphology. Sensory neurons were numbered and categorized into four types according to their morphology: type 1a (monociliated with collar), type 1b (monociliated without collar), type 1c (multiciliated without collar), and type 2 receptor cells (with non-emergent cilia) [3]. In total, the authors described 68 neurons. Forty neurons were considered sensory cells, while the type of non-sensory cells could not be identified (they could be motor neurons or interneurons) [3].The authors described 53 neurons concentrated in three pairs of ganglia (the frontal, ventral, and penis ganglia), while the remaining neurons were found in the periphery [3].The frontal ganglia consist of five paired cells A, B, C, D, and E (which may or may not be present) and cell F (which may or may not be paired); in addition, the perikarya of the sensory cells that form the anterior receptor group are part of the frontal ganglia (1–11,16–20) [3]. The ventral ganglia include paired cells H and I, which are located posteriorly [3]. Two penis ganglia are located ventrolateral to the copulatory organ [3]. Four non-sensory neurons—L, M, N, and O—are located in the periphery of these ganglia [3].

Data on the content of transmitters are very limited. Immunochemical studies have revealed only the posterior part of the organization of the nervous system, which is mainly involved in penial innervation [3]. The circumpenial nerve ring is serotonin-(5-HT), FMRFamide-, FVRIamide- and MIP-positive; some positive cells were detected near the circumpenial nerve ring in a previous study [13].

Data on the development of the nervous system of female *D. vorticoides* and *D. gyrociliatus* were recently described in detail [12]. The first neurons, which could be detected only with tubulin antibodies, are located at the anterior and posterior ends. Then, nerve development continues from both sides. Specific 5-HT and FMRFamide nerve elements appear later in development [12]. The same developmental pattern from both sides has been described for Errantia [10] and Sedentaria [11]. Oweniidae [5] and Sipunculida [7] show an anterior to posterior pattern of nervous system development. Nothing is known about the development of the nervous system of the male dwarf of *D. gyrociliatus.*

For this reason, the immunochemical study of mature dwarf males is required. However, the dwarf male is challenging to work with due to its size. The high loss of specimens during dissection, due to the miniscule size of the males (28µm×40µm), limits the success of these studies.

The aim of this work was to provide an up-to-date and comprehensive detailing of the morphology of the nervous system using immunochemical labeling. Here, we present data on the distribution of cilia, sensory neurons, and the 5-HT-like immunoreactive system in 3D reconstructions and cross-sections and compare our data with those of *D. gyrociliatus* females and previous results on metazoan dwarf males. These data will provide a better understanding of the evolution of the nervous system in Dinophiliformia and Annelida in general.

## 2. Materials and Methods

### 2.1. Animal Handling

The *Dimorphilus gyrociliatus* culture used in this study has been maintained since 2007. The animals were reared in 200 mL plastic tanks with artificial seawater (33‰ salinity) at 21 °C and fed with homogenized frozen nettle leaves (*Urtica* sp.) once every 7 days during water and tank changes. In this work, we studied mature *D. gyrociliatus* males only. To obtain enough male specimens, cocoons containing mature males (5 days after oviposition) were carefully removed from the tanks using a glass Pasteur pipette.

### 2.2. Fixation and Immunostaining

The samples were fixed with 4% paraformaldehyde (PFA) in phosphate-buffered saline (PBS, 0.01mM, pH=7.4) at 4 °C overnight. Then, the samples were subjected to immunostaining according to the protocol described in [12]. Each staining was performed with at least 50–60 male specimens. After fixation, the specimens were washed three times in PBS and then incubated for half an hour at room temperature in 10% normal goat serum and 1% bovine serum albumin in PBS. Subsequently, the samples were incubated at 10 °C for 3 days in a solution containing primary antibodies. A combination of anti-acetylated α-tubulin antibodies and anti-serotonin (5-HT) antibodies (Sigma-Aldrich, MO, USA; T-6793; mouse; monoclonal: clone 6-11B-1, ascites fluid and Immunostar, Hudson, WI, USA; 428002; rabbit; polyclonal; Product ID: 20080) diluted 1:5000–1:10,000 and 1:2000 in PBS with 0.1% Triton X-100 (PBS-TX) was used to label stabilized microtubules in ciliary bands and 5-HT-like immunoreactive elements, respectively. The primary antibodies were washed three times with PBS-TX solution and labeled with secondary goat anti-rabbit and goat anti-mouse antibodies with Alexa-555 (1:1000, Invitrogen, Waltham, MA USA; A-21428; goat; polyclonal) and Alexa-633 (1:1000, USA; A-21050; goat; polyclonal, Invitrogen, Waltham, MA USA) and phalloidin conjugated with Alexa-488 (A12379, Invitrogen, Waltham, MA USA), respectively, in PBS containing 0.1% Triton X-100 overnight at 10 °C. The primary antibodies were then rinsed with PBS. A DAPI stain (0.25 µg/mL) was used for the second rinse. The samples were then mounted on slides in 90% glycerol.

### 2.3. Microscopy and Image Processing

A Zeiss LSM-880 confocal scanning microscope (Karl Zeiss, Jena, Germany) was used to analyze the specimens. Optical stacks were acquired with a x63 objective and processed at 0.2 μm intervals using ZEN (Karl Zeiss, Jena, Germany) and Image J (NIH, Bethesda, Maryland, USA) to obtain two-dimensional images. The stacks were projected onto an image and then imported into Adobe Photoshop CC. Only the brightness and contrast were changed.

## 3. Results

Antibodies against acetylated α-tubulin label most neuronal cell bodies and their fibers. Cross-staining with antibodies against acetylated α-tubulin and the neurotransmitter serotonin (5-HT) allows for the visualization of specific neurons and their processes. Here, we provide a description of the external ciliation inner anatomy and neuronal pattern of the mature dwarf male of *Dimorphilus gyrociliatus.* The neural elements are described in the following order: type 1a, type 1b, type 1c and type 2 receptors; tubulin-positive nervous system; and 5-HT-like immunoreactive nervous system. All sensory neurons and structures are described using the classification invented by Windoffer and Westheide [3,4]. Individual neurons were recognized by determining their position according to the schematic diagram from Westheide and Windoffer.

### 3.1. External Ciliation

We observed three prominent ciliary structures on the body surface: an anterior ventral ciliary field at the anterior end of the ventral side (*avc*, Figure 3A–D); a posterior ventral ciliary field located posterior to the anterior ventral ciliary field on the ventral side (*pvc*, Figure 3A,B); and a dorsal ciliary field located at the anterior end of the dorsal side of the body (*dc*, Figure 3B–D).

There were several individual cilia belonging to sensory neurons. The most prominent were the cilia of the paired sensory neuron №9 (*sn9*, Figure 3B,C), which is located posterior to the dorsal ciliary field, and the cilia of sensory neurons №14 and №15, which are located at the posterior end of the body (*sn14* and *sn15*, Figure 3E). The other cilia belonged to sensory neurons located near the ciliary fields. Therefore, they may be visible on transverse sections.

### 3.2. Morphology

Two symmetrical seminal vesicles are located below the testis (*sv* and *test*, Figure 4A,B), and two symmetrical spermioducts connect the seminal vesicles to the copulatory organ (*spd*, Figure 4B). The copulatory organ is glandomuscular and consists of a muscular part and stylet gland cells (about 10 cells) (phalloidin and tubulin staining, *sgd*, Figure 4C,D). A pair of protonephridia is located symmetrically on the lateral sides of the testis (*, Figure 4A). There are also eight surface glands and one frontal gland.

### 3.3. Type 1a Cells

The two paired sensory neurons №1 (*sn1*, Figure 5A,A′,C) and sensory neurons №15 (*sn15*, Figure 5B,B′,D) were classified as type 1a receptors. All of these cells reacted positively with tubulin antibodies only. Both sensory neurons №1 and №15 showed no positive signal for 5-HT antibodies.

The two paired sensory neurons labeled №1 are located on the anterior surface over the anterior ciliary field (*sn1*, Figure 5A,A′).

The two paired sensory neurons labeled №15 are located on the posterior surface at the posterior end of the body (*sn15*, Figure 5B,B′). Each of these cells has a single cilium (*sn15*, Figure 5D).

### 3.4. Type 1b Cells

Nine paired sensory neurons are symmetrically located on the left and right sides of the body. Sensory neurons №2, 4 and 5 are located in the anterior part of the body and organized into an anterior receptor group (*sn2*, *sn4* and *sn5*, Figure 6A,A′,F,G).

The paired sensory neurons №7 are located near the dorsal ciliary field symmetrically near *E11* cells on the left and right sides (*sn7*, Figure 6A,A′,E). The paired sensory neurons №8 are located near the anterior ventral ciliary field and slightly posteriorly toward the anterior receptor group (*sn8*, Figure 6B,B′,I). The paired sensory neurons № 9 are located on the dorsal side, posterior to the dorsal ciliary field (*sn9*, Figure 6A,A′,D). The paired sensory neurons №10 are located at the lateral edge of the anterior ciliary field (*sn10*, Figure 6C,C′,J), slightly posterior to sensory neuron №8. The paired sensory neurons №11 are located symmetrically on the lateral side of the body (*sn11*, Figure 6B,B′,H). Two sensory neurons №14 are located below the genital opening (*go*), at the posterior part of the body (*sn14*, Figure 6B). These cells did not show 5-HT-like immunoreactivity.

### 3.5. Type 1c Cells

The posterior sensory neurons №12 and № 13 are symmetrically located on the lateral sides near the posterior ciliary field (*sn12* and *sn13*, Figure 7C,C′,F,G). Sensory neuron №12 has three cilia per cell (*sn12*, Figure 7F), whereas sensory neuron №13 has four cilia (*sn13*, Figure 7G). None of these cells exhibited 5-HT-like immunoreactivity.

### 3.6. Type 2 Cells

Four paired sensory neurons №16–20 are symmetrically distributed in the anterior part of the body close to the anterior ventral ciliary field (*sn16–20*, Figure 8A,A′,B,B′,C–F). All these sensory neurons are monociliated. None of these sensory neurons were detected with 5-HT-antibodies.

### 3.7. Acetylated-Tubulin Positive Nerve Elements

The dwarf male exhibits a complex nervous system (Figure 9A–D). The anterior part of the body contains the six acetylated tubulin-positive cell bodies, located posterior to the dorsal cilia (arrowheads, Figure 9A,A′). Two of the paramedial neuronal cells have thin projections that protrude dorsally (arrowheads, Figure 9A,A′). An unpaired cell body is located on the dorsal side under sensory neurons №9 (arrowhead, Figure 10A,B). This cell showed a positive response to 5-HT antibodies also (see next subsection, Figure 10A,B). At the anterior end of the body, dorsal and ventral commissures can be seen (*dc* and *vc*, Figure 9B,B′); these commissures follow the paired ventral bundles (*vb*, Figure 9B,B′). The paired sensory neurons №17 have thin processes that meet the commissures (Figure 9B,B′). Two paired cell bodies are located below this level (Figure 9C,C′). These cells also showed a positive reaction to 5-HT antibodies (see next subsection Figure 10A,A′,B,B′). Their projections form an additional commissure (*c*, Figure 9C,C′). At the posterior end of the body, there is a pair of neurons at the lateral edges of the posterior ventral ciliary field (*arrowheads*, Figure 9D,D′). Their projections form an additional commissure (*c*, Figure 9D,D′).

### 3.8. 5-HT-like Immunoreactive Elements

Two paired ventral 5-HT-immunoreactive neurons are symmetrically located at the level of the anterior ventral ciliary field (*arrowheads*, Figure 10A,B,D and Figure 11A,F,F′). Nuclear staining confirmed the presence of a single nucleus in each 5-HT-like immunoreactive neuron (*arrowhead*, Figure 11F,F′). The processes of these cells construct the main structures of the nervous system: the anterior neuropile, ventral bundles, and two commissures (*c*, Figure 10A,A′,B,B′ and Figure 11A). A single unpaired dorsal 5-HT-like immunoreactive cell is found at the anterior end on the dorsal side of the body (Figure 10A,A′,B,B′ and Figure 11D–D”). This unpaired cell is in proximity to sensory neurons №9 (Figure 11D). Two processes that come from a single stem of this cell protrudes towards the ventral bundles. At the anterior end, several 5-HT immunoreactive processes meet sensory neurons №7, 9, 14, and 17 (*sn7*, *sn9*, *sn14*, *sn17*, Figure 11B,D,E,F), the locomotory multiciliated cells (*E10*,*E11*, Figure 11C), and the copulatory organ (*co*, Figure 10D,D′ and Figure 11G,H). All these processes originate from paired ventral 5-HT-like immunoreactive neurons (Figure 10A,B,D and Figure 11A,F,F′).

Overall, it appears that 5-HT-like immunoreactive elements represent a small part of the nervous system.

## 4. Discussion

We provided detailed data on the organization of the nervous system of *D. gyrociliatus* dwarf males in terms of 5-HT immunoreactivity and clarified the distribution of ciliary and sensory neurons. A previous study using SEM and TEM [3,4] showed the complex organization of the nervous system and the distribution of sensory neurons and their possible functions, but lacked data on transmitter content.

### 4.1. Ciliary Landmark

Our results on the organization of cilia are similar to those of a previous study using SEM and TEM microscopy [3,4]. Three prominent ciliary fields were revealed: the anterior ventral, posterior ventral, and dorsal ciliary fields. In addition, there may be some homology between the ciliary structures in male and female *D. gyrociliatus*. These suggestions are based on the position on the body in males and females. Firstly, possible homology exists between the female dorsal acrotroch and male dorsal cilia because both these ciliary structures are located at the anterior part of the body (Figure 12). There is also possible homology between the anterior ciliary zone on the ventral side of females and the anterior ventral cilia of males, as both structures have multiple rows of ciliary cells and are located at the same position on the body (Figure 12). Finally, homology may exist between the ventral ciliary field in females and posterior ventral ciliary field in males, as both structures are located posteriorly on the ventral side of the body (Figure 12).

### 4.2. Sensory Neurons and CNS and 5-HT-like Immunoreactive Nervous System

Our results on the distribution of sensory neurons were in agreement with Westheide and Windoffer’s studies [3,4]. Twenty paired sensory neurons distributed on the body surface were found and described. These data verify our knowledge that sensory neurons are not 5-HT-like immunoreactive; nevertheless, the processes of 5-HT-like immunoreactive cells contact with some of sensory neurons (*sn7, sn14, sn17*), locomotory ciliary fields, and the copulatory organ. This fact might indicate that 5-HT is involved in sensing, the control of ciliary locomotion, and copulation in males, which is common in other annelid groups [10,11,15].

In terms of CNS organization, we observed some differences from Westheide and Windoffer’s studies [3,4]. The results of this study revealed six tubulin-positive neuronal cell bodies that might represent a frontal ganglia (Figure 9 and Figure 13). The previous study described six paired neuronal cell bodies that belonged to a frontal ganglia (paired cells A–E, 12 cells in total). According to position and a comparison with TEM micrographs from [3], these six tubulin-positive neuronal cell bodies may be similar to the A,B, and D paired cells. No signs of cells C and E were found in this study. 

The unpaired 5-HT-like immunoreactive cell underlying paired sensory neurons №9 is similar to the F cell described in the previous study [3]. In all analyzed specimens, this 5-HT-like immunoreactive cell was unpaired. It may interact with paired sensory neurons№9. In the previous study, there were specimens with both paired and unpaired F cells [3].

Some sensory neurons (*sn16–20*, *type 2* sensory neurons) were observed to be part of the frontal ganglia of the dwarf male, in agreement with Windoffer and Westheide’s data [3].

The results of the present study revealed anterior dorsal and ventral commissures that corresponded with previous data. Paired ventral bundles revealed both with tubulin and 5-HT antibodies in this study were also described in a previous study using TEM. The main difference was the number of cells revealed in the ventral nervous system. The results of this study showed a symmetrically located paired group of 5-HT-like immunoreactive cells at the level of the posterior ventral ciliary field and a posterior symmetrically located neuronal cell body, revealed with tubulin antibodies only (Figure 13). The previous results described paired H and I cells as a ganglion at the level of the first commissure and a more posteriorly paired group of four cells (L, M, N and O cells) at the level of the circumpenial nerve ring. The results of this study did not reveal these four paired cells; however, the same cells were described in the research published by Kerbl and coauthors as FMRFa-, FVRI-, and MIP-positive cells [13].

Further research is needed to identify the signaling molecules expressed in the sensory neurons and central nervous system of the dwarf male. A recent study showed that the transition of the ventral nervous system into ladder-like, cluster-bearing ventral cords with a subepidermal position may have evolved independently in a few groups after the division of Pleistoannelida into Errantia and Sedentaria. A subepidermal ventral nervous system is considered as the derived condition within Annelida [16]. The configuration of the ventral nervous system for the annelid ground pattern therefore differs from the commonly accepted strict subepidermal ladder-like configuration (meaning somata only within the paired ganglia, paired somata-free connectives, and segmentally arranged commissures) [16].

An intraepidermal ventral nervous system has been observed within Palaoannelida, Chaetopteriformia, and several Pleistoannelida (Errantia and Siboglinidae from Sedentaria) (Figure 14). Scholars assume that the transition into this condition occurred several times in the course of annelid evolution [16]. The results of a recent study on Dinophiliformia demonstrated that in *D. vorticoides* and *D. gyrociliatus* females, the ventral nervous system is located subepidermally [14]. The present study on *D. gyrociliatus* males showed an intraepidermal ventral nervous system. This may indicate a push back towards the ancestral condition in the organization of males.

The details of *D. gyrociliatus* female neurogenesis showed that the first nerve elements appear at the anterior and posterior poles of the embryo. These elements are not 5-HT- or FMRFamide-positive and specific nerve elements appear later in development. 5-HT-like nerve elements appear at the middle trochophore stage as three cells located symmetrically on the left and right sides at the base of the first commissure [12]. The long basal processes of these cells join the first commissure and the ventral bundles going posteriorly and anteriorly to the brain neuropile. The organization of the dwarf male 5-HT-like immunoreactive system includes dorsal and ventral anterior commissures, paired ventrolateral bundles, ventral ganglia, and two ventral commissures. Thus, the pattern is similar to that of *D. gyrociliatus* females at their middle trochophore stage (Figure 15) [12].

### 4.3. Apical Organ

Annelid larvae possess a prominent apical organ [5,10,11,14,15,16,17,18,19,20] However, some exceptions are present; it may have been reduced in *C. teleta* [21,22,23], *D. vorticoides* [12], and *D. gyrociliatus* females [12]. At the same time, the apical organ is present in other representatives of Sedentaria [11] and Errantia [10,15].

The absence of the apical organ in *C. teleta*, *D. vorticoides,* and *D. gyrociliatus* females may be a result of independent evolutionary events.

However, a recent study on *D. vorticoides* and *D. gyrociliatus* demonstrated flask-shaped cells in the anterior region of *D. gyrociliatus* and *D. vorticoides* embryos that could be the rudiments of an apical organ [12]. Thus, the apical organ of Dinophiliformia may have been reduced or evolved in a way to use other signaling molecules, such as those highly expressed in the brain of the adult forms of Dinophiliformia and in the apical organ cells of other annelids [24].

The present study on the dwarf male revealed a group of three cells in the anterior region that were assigned to the miniaturized apical organ: paired sensory neurons №9 and an unpaired 5-HT-like immunoreactive cell (F) below these neurons (Figure 13 and Figure 15). Sensory neurons №9 demonstrated no 5-HT-like immunoreactivity; however, they may be innervated by the unpaired 5-HT-like immunoreactive F cell. This group of three cells is located at the anterior end on the dorsal side of the body. Unlike males, *D. gyrociliatus* females lack this group of cells; moreover, there is no such structure during embryonic development. 

### 4.4. Male Dwarfism

Male dwarfism is observed in several annelid groups: dinophilidae [3,4,13,14], siboglinidae [25,26], spionidae [27], and bonellid echiurans [28,29]. There are similar traits between the nervous system organization of *D. gyrociliarus* dwarf males and *Osedax* dwarf males in terms of the distribution pattern of the anterior neurons, anterior ventral and dorsal commissures, paired ventral and dorsal bundles, and paired groups of two perikaya [25]. As with *D. gyrociliatus*, most nerve elements of *Osedax* males were detected only with acetylated tubulin antibodies [25]. At the same time, in terms of size and external ciliation pattern, the *Osedax* male resembles the *D. gyrociliatus* female. Thus, both possess a prominent prototroch. 

Two possible mechanisms may be involved in the origin of dwarf males. Either heterochrony, represented by proportioned dwarfism or in the form of progenesis, or miniaturization [1,30] These two processes are reflected in the morphology of dwarf males, particularly involving the nervous system [14]. During progenesis, the dwarf male is suggested to be similar to the earlier developmental stage of normal-sized females. Dwarfs appearing as a result of miniaturization are different from normal-sized females; they lack many characteristics of normal-sized females and may have specific adaptations to the reduction. These adaptations may not have any homology to the structures of the normal-sized females [14]. The morphology of bonellid echiurans resembles larvae [25,26], while the morphology of dwarfs in *D.gyrociliatus* [14] and *Scolelepis laonicola* (Spionidae) is different from the normal-sized females [27].Thus, they may have originated through a combination of heterochronous and non-heterochronous evolutionary paths [12,27].

The results of the present study revealed some similar characteristics between *D. gyrociliatus* males and females in terms of external ciliary pattern and the 5-HT-like immunoreactive system. These similarities may indicate progeny as the origin of the *D. gyrociliatus* dwarf male.

Among non-annelid groups, the *D. gyrociliatus* dwarf male demonstrates a strong similarity with *Symbion pandora* (Cycliophora) dwarf males. They are the same size and possess similar external ciliary fields. The dorsal ciliary field and anterior ventral ciliary field in *D. gyrociliatus* male are similar to the frontal cilia and ventral ciliated field of *S. pandora* males, respectively [31]. This similarity may be a result of independent evolutionary events in Dinophiliformia and Cycliophora.

## 5. Conclusions

The present study revealed that dwarf males may have similar traits to females in terms of external ciliation and nervous system organization. Thus, the ciliary structures may be homologous to those of *D. gyrociliatus* females. The nervous system of *D. gyrociliatus* dwarf males is located intraepidermally, with some sensory frontal ganglia neurons located on the body surface. 5-HT-like immunoreactive elements compose only a minor part of the nervous system. None of the revealed sensory neurons demonstrated 5-HT-like immunoreactivity. However, 5-HT-immunoreactive processes were found to contact the anterior part of sensory neurons, locomotory cilia, and the copulatory organ, which might indicate the involvement of 5-HT in sensing, ciliary locomotion, and copulation. The organization of the nervous system in *D. gyrociliatus* males is similar to that of the female during the middle trochophore stage. In addition, *D. gyrociliatus* dwarf males may exhibit a compact apical organ made up of two sensory neurons and one 5-HT-like immunoreactive cell. The fact that *D. gyrociliatus* males, unlikefemales, retain this apical organ reflects an extreme degree of pedomorphosis in males.

## Figures and Tables

**Figure 1 biology-11-01674-f001:**
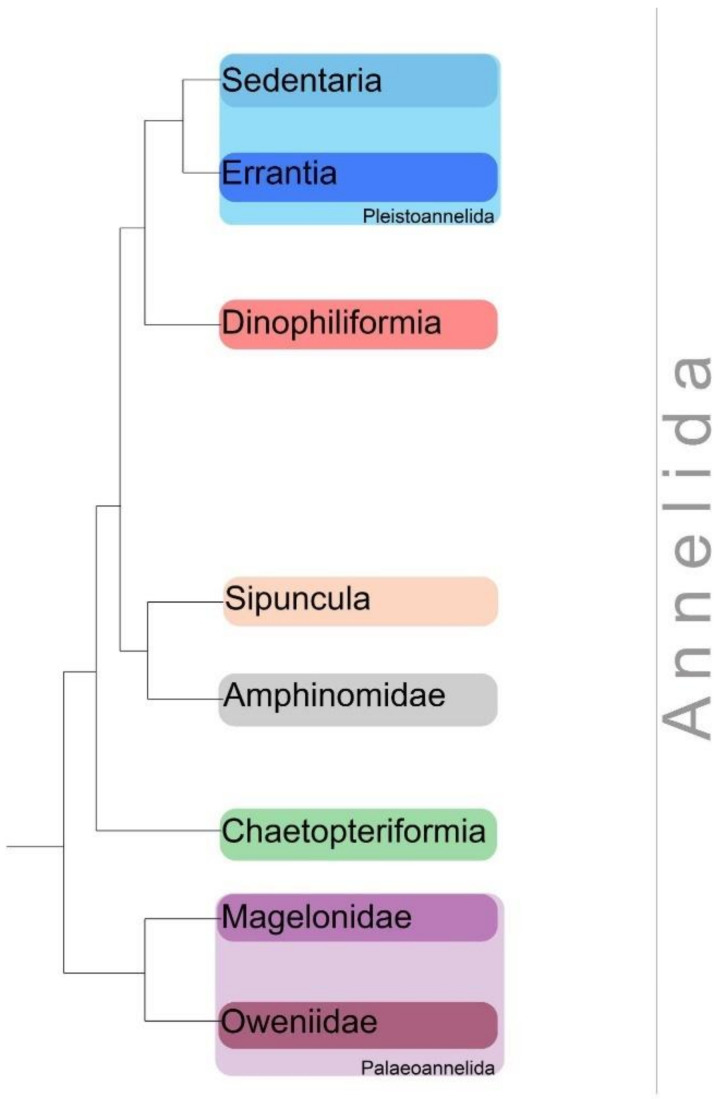
Annelida phylogenetic tree.The diagram is based on recent data. The phylogeny was retrieved from [1,2]. Dinophiliformia is a newly emerged group, sister to Pleistoannelida.

**Figure 2 biology-11-01674-f002:**
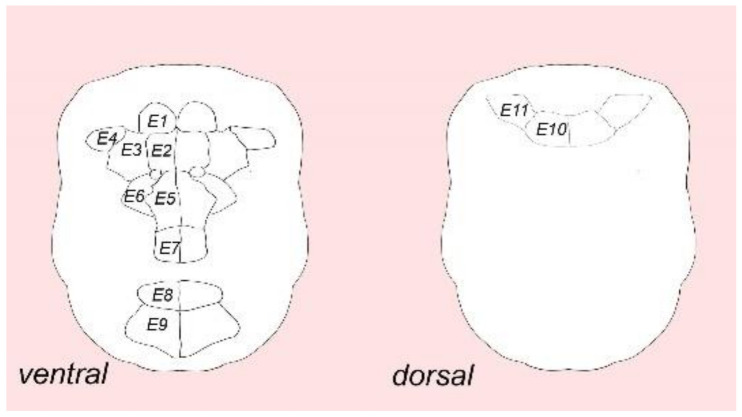
Multiciliated cells of ciliary fields of *Dimorphilus gyrociliatus* dwarf male. The diagram is based on the results presented in [3,4]. The multiciliated cells distriburion pattern is adapted from [3,4] Copyright © 1988 Alan R. Liss, Inc. Apical end is up, relative size is not to scale. *E1–E11* represent multiciliated cells of anterior ventral, posterior ventral and dorsal ciliary fields.

**Figure 3 biology-11-01674-f003:**
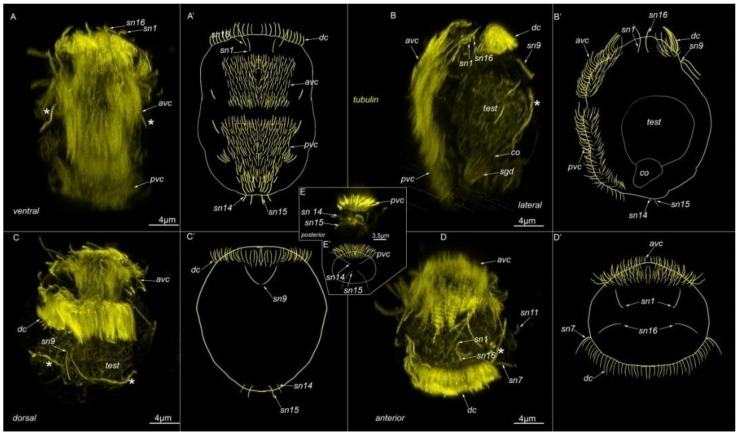
External ciliation in mature *Dimorphilus gyrociliatus* dwarf male revealed by anti-acetylated alpha-tubulin labeling. On (**A**–**C**) apical end is up. (**A**,**A′**) Ventral view on wholemount. At the anterior part of the body, there is a prominent anterior ventral ciliary field (*avc*). The posterior ciliary field is located posteriorly the anterior ventral ciliary field. A pair of protonephridia (*) is located symmetrically on the body. At the anterior end prominent cilia of sensory neurons №1 and №16 are visible. (**B**,**B′**). Lateral view on wholemount. At the anterior part of the body, the dorsal ciliary field (*dc*) is present. Posteriorly, two single cilia of sensory neurons №9 are located. A prominent testis (*test*) fills the most of the inner part of the body. Posteriorly to testis copulatory organ with multiple stylet glad cells (*sgd*) are located. (**C**,**C′**). Dorsal view on wholemount. (**D**,**D′**). Anterior view of the wholemount. (**E**,**E′**). The posterior surface of wholemount. At the posterior tip cilia of paired sensory neurons №14 and №15 are located. Abbreviations: *avc*—anterior cilia; *dc*—dorsal cilia; *pvc*—posterior cilia; *co*—copulative organ; *sgd*—stylet gland cell; *sn1*–*sn20*—sensory neurons 1–20; *test*—testis; *—nephridia.

**Figure 4 biology-11-01674-f004:**
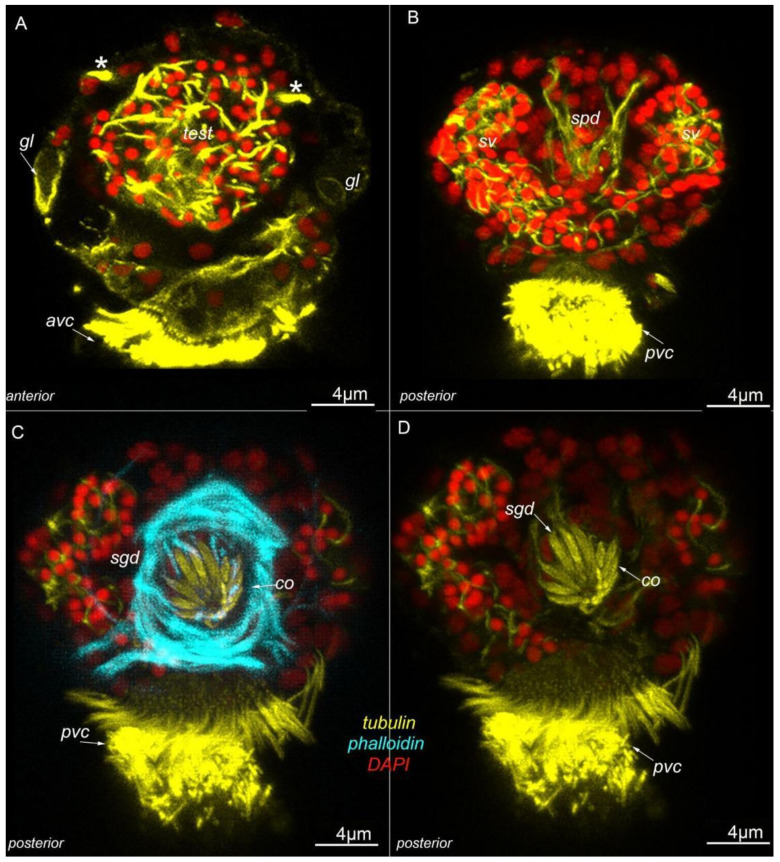
The internal organization of *Dimorphilus gyrociliatus* dwarf male. (**A**–**D**) optical cross-sections of the anterior (**A**,**B**) and posterior (**C**,**D**) parts of the body. (**A**) Prominent testis (*test*) and a pair of protonephridia (*), a pair of surface gland cells (*gl*). (**B**) symmetrically located seminal vesicles (*sv*) and unpaired spermioduct (*spd*). (**C**) Copulatory organ—glandomuscular organ, includes glandular part, represented by stylet gland cells (*sgd*) and muscle part (cyan). (**D**) The same section as (**C**); without cyan channel, thus only stylet gland cells are visible, without phalloidin.

**Figure 5 biology-11-01674-f005:**
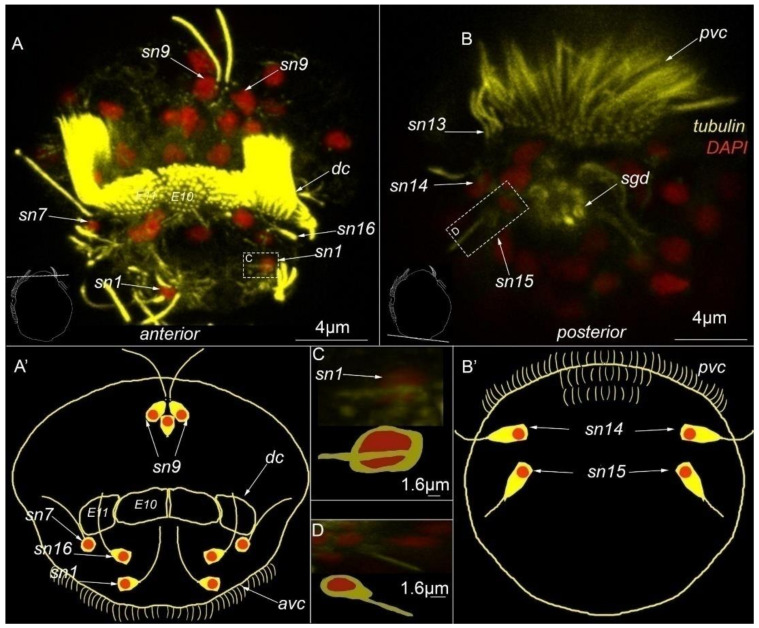
Type 1a sensory neurons in mature *Dimorphilus gyrociliatus* dwarf male revealed by anti-acetylated alpha-tubulin labeling. (**A**,**A′**) Anterior surface view. (**A′**) Schematic representation of the anterior surface. (**B**) The posterior surface view. (**B′**) Schematic representation of the posterior surface view. (**C**) The close-up and drawing of sensory neuron №1. (**D**) Close-up and drawing of sensory neuron №2. Abbreviations: *E10, 11*—multiciliated cells of the dorsal ciliary field; *dc*—dorsal cilia; *pvc*—posterior cilia; *sgd*—stylet gland cell; *sn1*–*sn20*—sensory neurons 1–20.

**Figure 6 biology-11-01674-f006:**
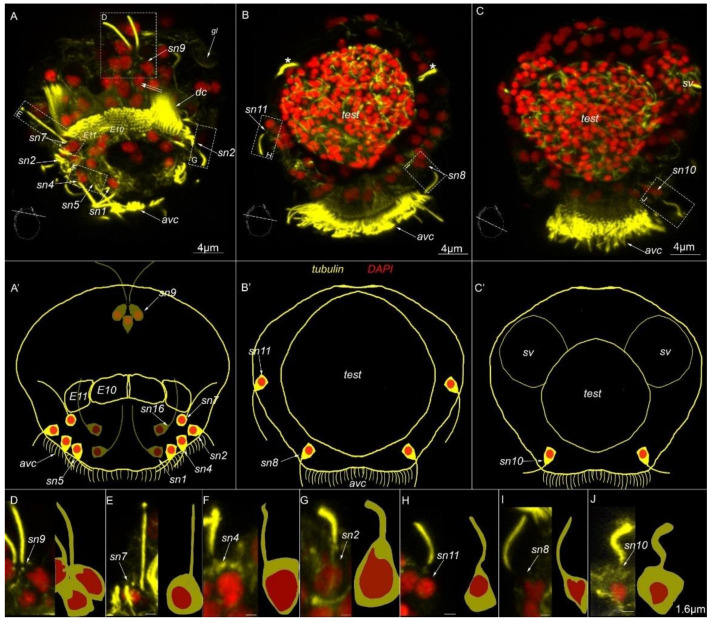
Type 1b sensory neurons in mature *Dimorphilus gyrociliatus* dwarf male revealed by anti-acetylated alpha-tubulin labeling. (**A**,**A′**) A cross-section and schematic representation of the anterior surface. (**B**,**B′**) A cross-section and schematic representation of the anterior part of the body. (**C**,**C′**) A cross-section and schematic representation of the middle part of the body. (**D**) The close-up and drawing of sensory neuron №9. (**E**) The close-up and drawing of sensory neuron №7. (**F**) The close-up and drawing of sensory neuron №4. (**G**) The close-up and drawing of sensory neuron №2. (**H**) The close-up and drawing of sensory neuron №11. (**I**) The close-up and drawing of sensory neuron №8. (**J**) The close-up and drawing of sensory neuron №10. Abbreviations: *avc*—anterior cilia; *E10,11*—multiciliated cells of the dorsal ciliary field; *dc*—dorsal cilia; *pvc*—posterior cilia; *test*—testis; *sv*—seminal vesicles; *sn1*–*sn20*—sensory neurons1–20; *—nephridia.

**Figure 7 biology-11-01674-f007:**
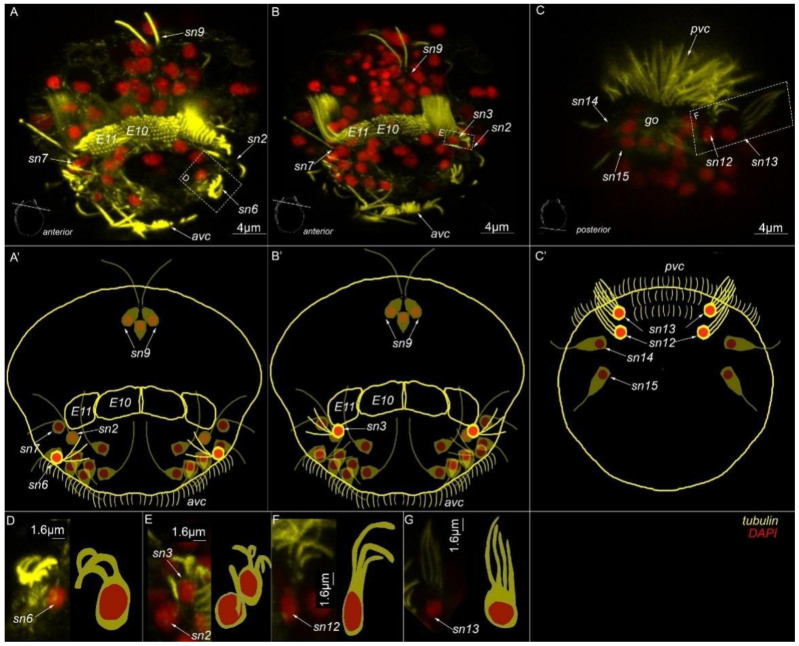
Type 1c sensory neurons in mature *Dimorphilus gyrociliatus* dwarf male revealed by anti-acetylated alpha-tubulin labeling. (**A**) Anterior surface view. (**A′**) Schematic representation of the anterior surface. (**B**) Anterior surface view. (**B′**) Schematic representation of the anterior surface. (**C**) The posterior surface view. (**C′**) Schematic representation of the posterior surface view. (**D**) The close-up and drawing of sensory neuron №6. (**E**) Close-up and drawing of sensory neuron №3. (**F**) Clode-up and schematic drawing of sensory neuron №12. (**G**) Close-up and schematic drawing og sensory neuron №13. Abbreviations: *E10, 11*—multiciliated cells of the dorsal ciliary field; *dc*—dorsal cilia; *pvc*—posterior cilia; *sgd*—stylet gland cell; *sn1–sn20*—sensory neurons 1–20.

**Figure 8 biology-11-01674-f008:**
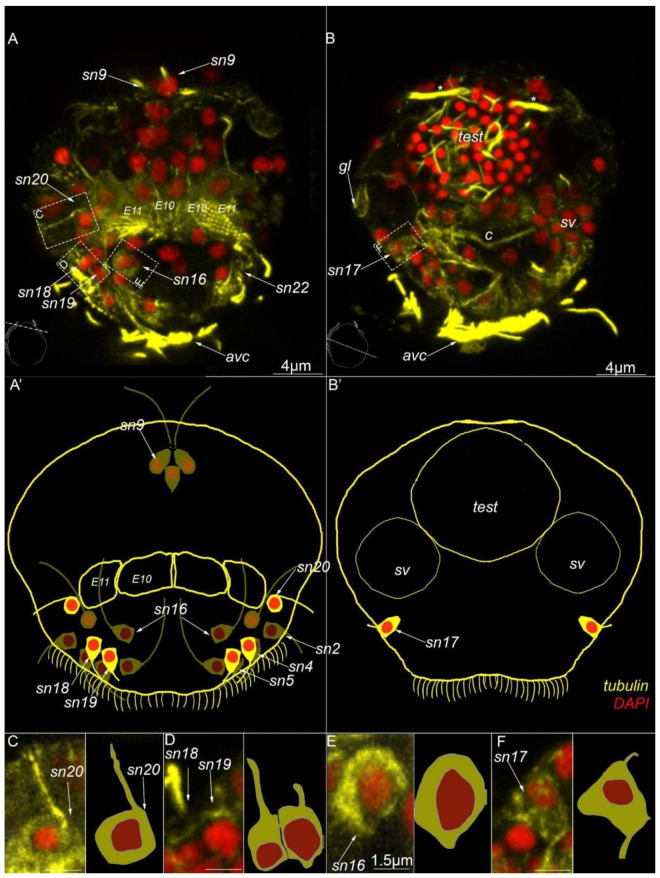
Type 2 sensory neurons in mature *Dimorphilus gyrociliatus* dwarf male revealed by anti-acetylated alpha-tubulin labeling. (**A**) Anterior surface view. (**A′**) Schematic representation of the anterior surface. (**B**) The posterior surface view. (**B′**) Schematic representation of the posterior surface view. (**C**) The close-up and drawing of sensory neuron №20. (**D**) Close-up and drawing of sensory neurons №18 and 19. (**E**) The close-up of sensory neuron №16. (**F**) The close-up of sensory neuron № 17. Abbreviations: *E10, 11*—multiciliated cells of the dorsal ciliary field; *dc*—dorsal cilia; *pvc*—posterior cilia; *sgd*—stylet gland cell; *sn1–sn20*—sensory neurons 1–20.

**Figure 9 biology-11-01674-f009:**
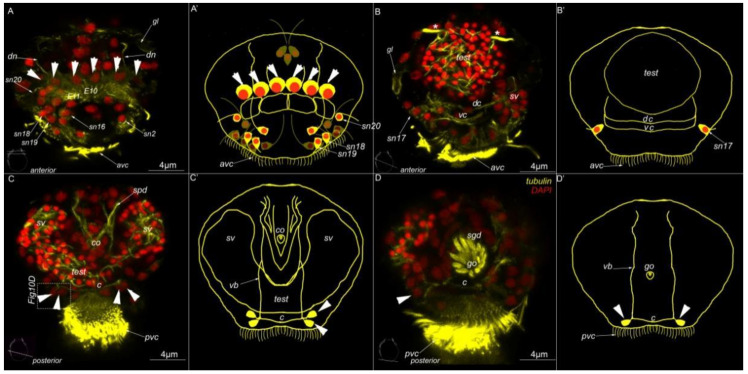
Tubulin-like immunoreactivity in the mature dwarf male of *Dimoprhilus gyrociliatus*. (**A**,**A′**) Anterior view on wholemount and schematic representation. (**B**,**B′**) A cross-section at the middle part of the body and its schematic representation. (**C**,**C′**) A cross section of the posterior body region and its schematic representation. (**D**,**D′**) A cross section of the posterior body region at the level of stylet gland cells and its schematic representation. Abbreviations: arrowheads indicate neuronal cell bodies; *avc*—anterior ventral cilia; *dc*—dorsal commissure; *vc*—ventral commissure; *c*—commissure; *co*—copulatory organ; *go*—genital opening; *test*—testis; *pvc*—posterior cilia; *sgd*—stylet gland cell; *sn1–sn20*—sensory neurons1–20; *sv*—seminal vesicles; *vb*—ventral bundle; *—nephridia.

**Figure 10 biology-11-01674-f010:**
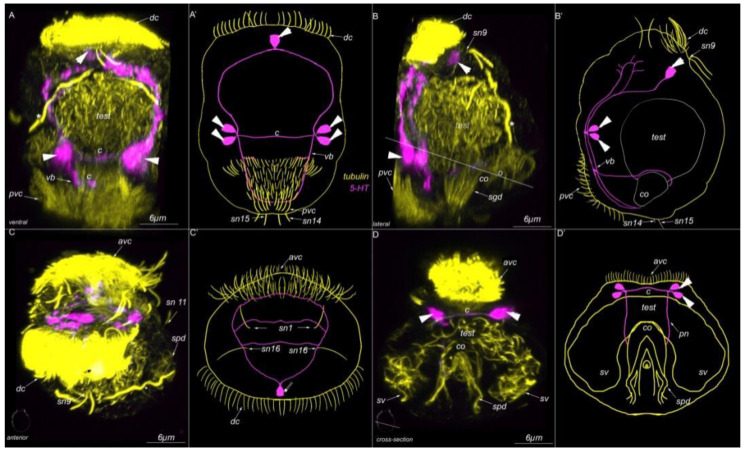
5-HT-like immunoreactivity in the mature dwarf male of *Dimorphilus gyrociliatus.* (**A**,**A′**) Ventral view on wholemount. (**B**,**B′**) lateral view on wholemount. (**C**,**C′**) dorsal view on wholemount. (**D**,**D′**) A cross section of the posterior body region as indicated in (**C**,**C′**) Abbreviations: arrowheads indicate neuronal cell bodies; *avc*—anterior ventral cilia; *c*—commissure, *dc*—dorsal cilia; *test*—testis, *pn*—penial nerve; *pvc*—posterior cilia; *sgd*—stylet gland cell; *spd*—spermioduct; *sn1–sn20*—sensory neurons1–20; *sv*—seminal vesicles; *vb*—ventral bundle; *—nephridia.

**Figure 11 biology-11-01674-f011:**
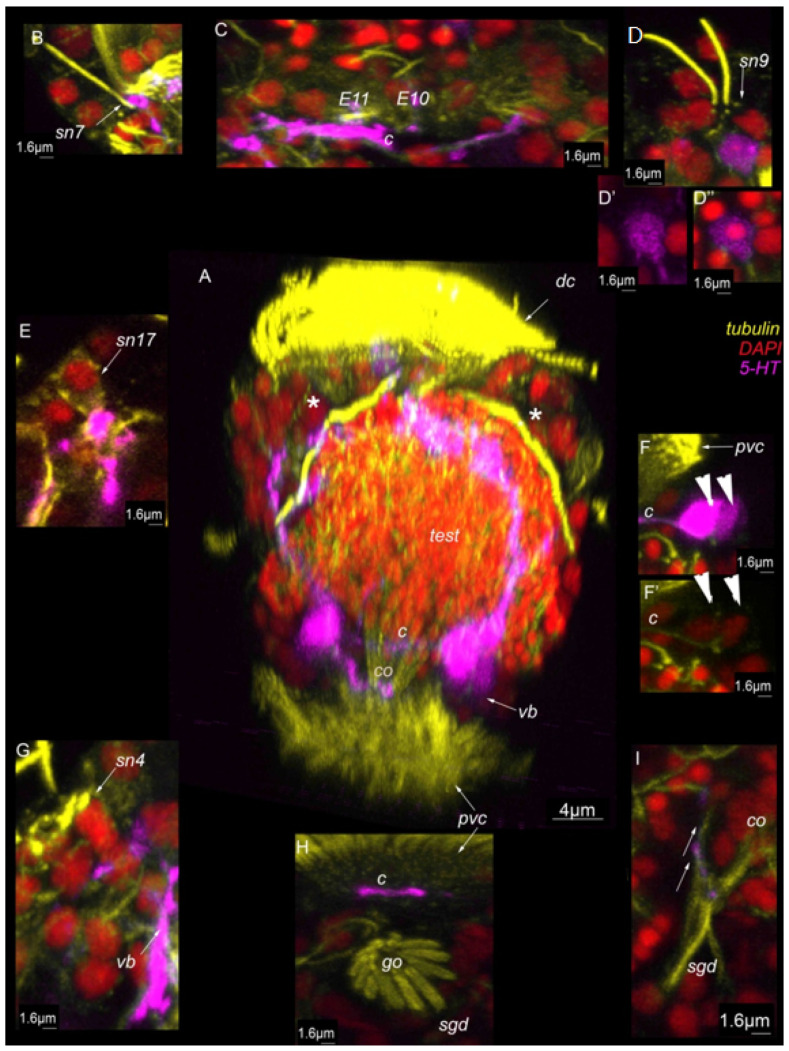
5-HT-like immunoreactivity in the mature dwarf male of *Dimoprhilus gyrociliatus*. (**A**)Ventral view on wholemount. (**B**) Sensory neuron№7 and thin 5-HT-like immunoreactive processes. (**C**) Multiciliated cells of dorsal ciliary field are innervated by 5-HT-like immunoreactive processes. (**D**,**D′,D″**) Sensory neurons №9 and unpaired 5-HT-like immunoreactive cell. (**E**) Sensory neuron №17. (**F,F′**) The cell bodies of 5-HT-like immunoreactive cells. (**G**) Sensory neuron №4. (**H**) Commissure. (**I**) 5-HT-like immunoreactive processes innervate copulatory organ. Abbreviations: arrowheads indicate neuronal cell bodies; *avc*—anterior ventral cilia; *c*—commissure; *dc*—dorsal cilia; *test*—testis; *pvc*—posterior cilia; *sgd*—stylet gland cell; *sn1–sn20*—sensory neurons 1–20; *sv*—seminal vesicles; *—nephridia.

**Figure 12 biology-11-01674-f012:**
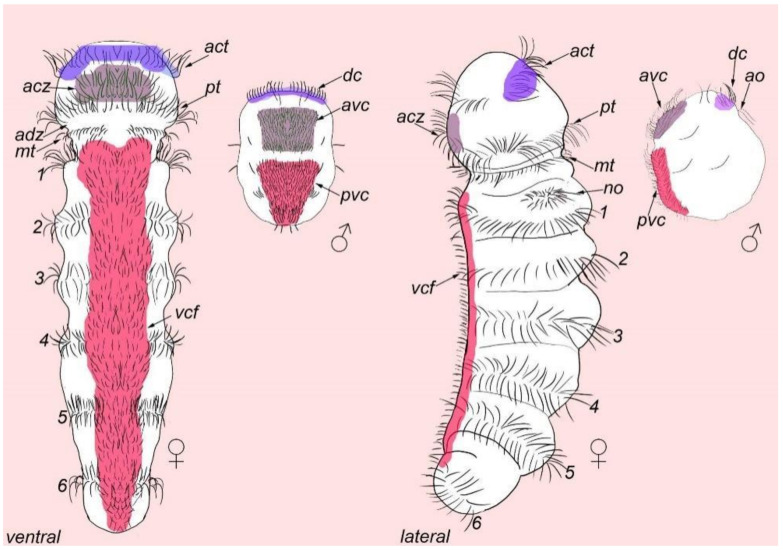
Comparative diagram of the external ciliation and possible homology regions on *D. gyrociliatus* male and female. The diagram is based on recent data on *D. gyrociliatus* female [14] and male (this study) external ciliation and male. Female acrotroch and male dorsal cilia, anterior ciliary field in female ad anterior ventral cilia in male, ventral ciliary field in female and posterior ventral cilia in male seems to be homologus structures. The apical end is up. Relative dimensions are not to scale. Abbreviations: *act*—acrotroch; *acz*—anterior ciliary field; *adz*—adoral ciliary zone; *avc*—anterior cilia; *dc*—dorsal cilia; *mt*—metatroch; *pvc*—posterior cilia; *test*—testis; *co*—copulatory organ.

**Figure 13 biology-11-01674-f013:**
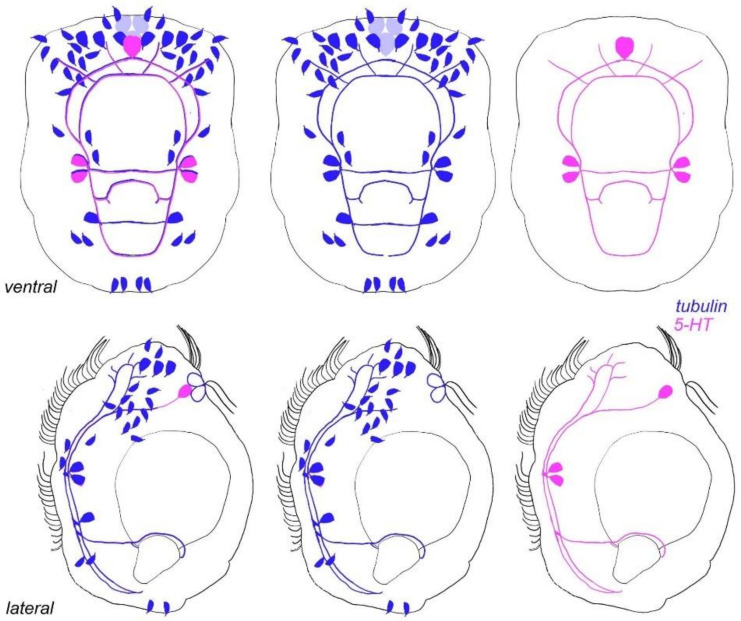
Schematic representation of nervous system morphology in *D. gyrociliatus*. Apical end is up. Tubulin–blue; 5-HT–magenta. 5-HT-like immunoreactive elements represent a minor part of the whole nervous system.

**Figure 14 biology-11-01674-f014:**
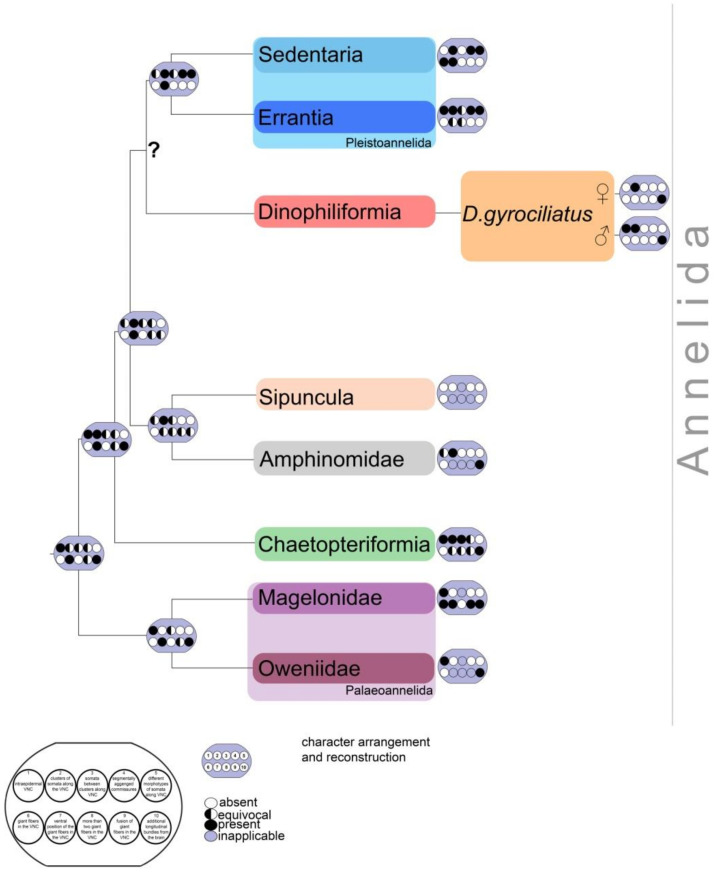
Schematic diagram of the ventral nervous system anatomy across Annelida phylogenetic tree with addition of *Dimorphilus gyrociliatus* data. Phylogenetic tree is based on [1,2] the characters pattern and distribution is based on [16]. The legend and character arrangement and reconstruction matrixes were adapted from [16]. The characters are coded in a way: absent (white), present (black) and inapplicable (grey). Note the difference between *D. gyrociliatus* male and female pattern.

**Figure 15 biology-11-01674-f015:**
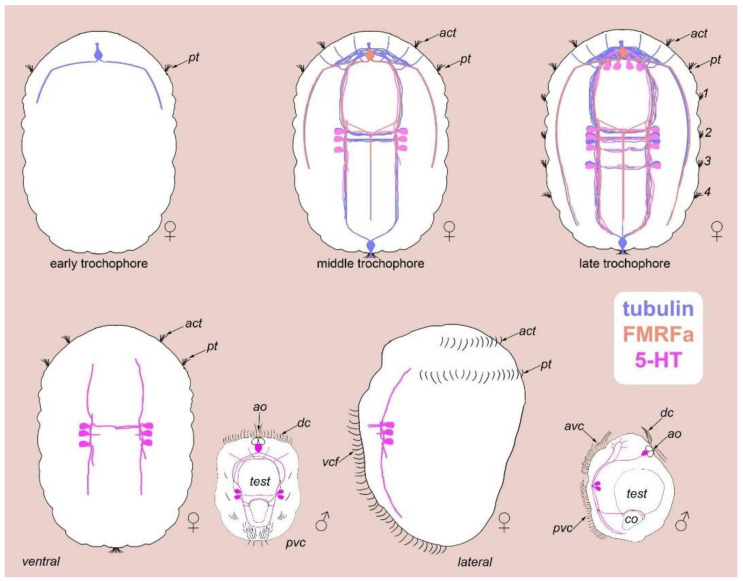
Schematic representation of *D. gyrociliatus* females nerve development and comparison with dwarf male. The diagram is based on [12]. Females go through three developmental stages: early trochophore, middle trochophore, and late trochophore. The dwarf male 5-HT-like immunoreactive system is similar to the female nervous system during the middle trochophore stage.

## Data Availability

Not applicable.

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
