# Peer review of "Dimorphilus gyrociliatus (Annelida: Dinophiliformia) Dwarf Male Nervous System Represents a Common Pattern for Lophotrochozoa"

_biology, 2022, doi:10.3390/biology11111674_

Round 1
Reviewer 1 Report
general comments: The article describes the nervous system components of D. gyrociliatus dwarf males in great detail and sufficiently using different antibodies.
Overall, the presentation of the figures can be improved, as it is sometimes confusing and not easy to understand. For example, original pictures and schematic drawings of the same should be positioned close together. Separation lines would also help the reader to better locate the individual panels. All figures should include basic information, like which colors correspond to which staining, or male and female association - so far this is only present in some figures. All unnecessary information should be removed from the image or explained - What is blue in Fig. 5?
The method section is lacking some information on 5-HT, but provides information on Phalloidin, which is never mentioned nor shown in any figure.
The discussion about the ASO is a bit weak. No real or schematics are shown to demonstrate that a homology between both structures can be claimed. This should be improved, especially since the alleged homology is one of the main findings of the paper. The text in papragraph 4.4 is referencing Fig 9, but that doesn't depict what is described.
specific comments: lines 74-77: Sentence is not understandable, please re-phrase
lines 94 & 307: genus names are not in italics
Throughout the text there are inconsistencies regarding the way of numbering - 3 different styles are used - please pick one and use throughout.
The same is true of the text style/font used. It varies between Fig. legends, but should be the same throughout.
Fig. legend 7 is incomplete.
The color used in Fig 9 for anterior and dorsal are too similar to be easily distinguished.
Author Response
Point 1: general comments: The article describes the nervous system components of D. gyrociliatus dwarf males in great detail and sufficiently using different antibodies.
Overall, the presentation of the figures can be improved, as it is sometimes confusing and not easy to understand. For example, original pictures and schematic drawings of the same should be positioned close together. Separation lines would also help the reader to better locate the individual panels. All figures should include basic information, like which colors correspond to which staining, or male and female association - so far this is only present in some figures. All unnecessary information should be removed from the image or explained - What is blue in Fig. 5?
Response 1: Thank you for comment. I corrected the figures Pictures and schematic drawings are now close together, I also added separation lines and checked basic information which color correspond to which staining: yellow -tubulin, red- DAPI and magenta- 5-HT. Blue in Fig. 5 is Phalloidin, I removed it from the plate.
Point 2:The method section is lacking some information on 5-HT, but provides information on Phalloidin, which is never mentioned nor shown in any figure.
Response 2: Thank you to mention about 5-HT antibodies: 1:2000 dilution , Immunostar, Hudson, WI, USA; 428002; rabbit; polyclonal; Product ID: 20080 . I included this information into method section.
Point 3: The discussion about the ASO is a bit weak. No real or schematics are shown to demonstrate that a homology between both structures can be claimed. This should be improved, especially since the alleged homology is one of the main findings of the paper. The text in papragraph 4.4 is referencing Fig 9, but that doesn't depict what is described.
Response 3: I've drastically changed the introduction, results and discussion. I added schemes and more details into discussion section.
Point 4: specific comments: lines 74-77: Sentence is not understandable, please re-phrase
Response 4: I' ve rewritten the Introduction section
Point 5: lines 94 & 307: genus names are not in italics
Response 5: corrected
Point 6: Throughout the text there are inconsistencies regarding the way of numbering - 3 different styles are used - please pick one and use throughout.
Response 6: Thank you, I've chosen sn1-20, like in figures
Point 7: The same is true of the text style/font used. It varies between Fig. legends, but should be the same throughout.
Response 7:I agree. I verified several times font and text style, using template.
Point 8: Fig. legend 7 is incomplete.
Response 8: Thank you. I added information into Fig.7 legend.
Point 9: The color used in Fig 9 for anterior and dorsal are too similar to be easily distinguished.
Response 9:I changed the color.

Reviewer 2 Report
Overview
The author describes morphology of the ciliary bands and nervous system in the dwarf male of the meiofaunal, paedomorphic species Dimorphilus gyrociliatus using anti-serotonin and anti-acetylated tubulin immunolabeling and confocal microscopy. Previous molecular works describing the adult nervous system and ciliary bands in D. gyrociliatus have largely focused on the females of this species. Overall, the data in this paper are an important contribution to our understanding of lophotrochozoan ciliary band evolution and the nervous system structure in an interstitial, paedomorphic annelid. However, as written, the Results section lacks sufficient explanation of the findings. Furthermore, comparisons with results for D. gyrociliatus dwarf males from other papers (e.g. Windoffer and Westheide 1988), females of the same species, and with other annelid species, both miniaturized and not, are needed to put this work in context and elucidate its significance and potential evolutionary implications.
Major comments
The paper lacks detail in many areas (Intro, Results and Discussion), and the descriptions are fairly precursory. The figure panels are well-labeled with many different views, but having a clearer, written description of the data is necessary. I also think a better explanation of what was known about the morphology of the dwarf males prior to this study and how this work compares with other work (see above) is necessary.
It is difficult for me to tell why all of the single, ciliated cells are considered sensory neurons. I do not see neurites or other morphology that would indicate a neuronal cell. Also, some explanation of the classification system being used would be helpful. For example, in lines 162 – 165, what makes these “type 1a cells”? These cells do not look different than the “type 1b cells” to me, so an explanation of how they were classified seems important. Is this based on specific morphological features?
Minor comments
Introduction
lines 74 - 76 I think there is a verb missing in this sentence.
“In addition, during development, the prototroch nerve….”
lines 78 – 79 This sentence needs to be re-written. I am not certain what you mean by this statement. Are you saying that no ASO has been found within species belonging to Dinophiliformia? Or that no ASO has been visualized in females of D. gyrociliatus?
“D. gyrociliatus belongs to the lophotrochozoan Annelida, in whose representatives no ASO was mentioned.”
Lines 80 – 81 Please add a few sentences about the morphology of the dwarf males. For example (mentioned below), it would be helpful to know if these dwarf males have a stomadeum and/or gut.
Methods
I could not find the clone and dilution for the anti-5HT antibody. I also could not find the dilution used for phalloidin. How were the immunolabeled specimens stored before imaging? Were they stored in the 90% glycerol (in PBS?) at 10 oC before imaging?
Results
In general, within the text, I would refer to the abbreviations for the structures along with the Figure panels. For example, the first time that you mention the “anterior ciliary field” line 147, I would refer to (avc, Figure 1 A–D). Also, I might provide a little more description of the ciliary fields. Later in the text you mention multi-ciliated cells that I think are part of the ciliary bands, but I wasn’t certain how this differed from the multi-ciliated cells that were labeled as “type 1c”.
line 154 I think you mean that the posterior ciliary field is located posterior to (rather than “below”) the anterior ciliary field. Also, from the images and diagrams, it looks like both the avc and the pvc are localized to the ventral face of the animal. Is this correct? If so, I would mention this in the text. I might also add “ventral” to the names for both.
line 161-162 Add a citation.
Keep the same nomenclature for the sensory neurons throughout. Sometimes they are referred to as “sensory neuron no. X” and then in the same sentence “sensory neuron # X”
Line 234 and Figure 7 What is meant by “tubulin-like”? For the previous figures, cells were referred to as ‘labeled with anti-acetylated tubulin’. Usually, authors use the term “-like” for cross-reactive antibodies that may be labeling something other than their original antigen. But I do not see why the elements being described in this section are not just acetylated-tubulin positive. I just was not certain if this change in wording was meant to indicate some difference or not.
Lines 252 – 253 Why are these two pairs of 5HT+ cells considered “ganglia” and not neurons associated with the stomodeum as in other annelids? Do the dwarf males have a mouth and/or gut?
Figure 1 It’s difficult to see panel “E”. I might make the white line around this inset a little thicker. For D, I am not certain what view is being shown (i.e. does “top view” refer to an “anterior view”). For the bottom panel of diagrams, I would add a horizontal line between these and the panels above so that it’s clear the “dorsal” and “top” labels associated with C and D, respectively, are separate from the diagrams. Otherwise, the diagrams and labels in this figure are very helpful.
Figures 2, 4 and 5 Why are the multi-ciliated cells referred to as “E10” and “E11”? This nomenclature is only used in the figure legends but is not described anywhere in the text that I can find.
Figure 5 What is the cyan labeling in panel A?
Figures 7 and 8 Add an explanation of what each color is showing (i.e. magenta is anti-5HT, red is…etc). Also, what does “co” refer to (copulatory organ)?
Discussion
Line 284 A better explanation of the results from the Windoffer and Westheide 1988 paper in the Discussion would be helpful. It is not very clear to me how your results compare to theirs.
Lines 290 – 291 I do not understand this statement as written. Why do the two sexes undergo ‘the same developmental program’? Why couldn’t there be initiation of a sex-specific developmental program that generated a non-homologous ciliary band in one but not the other sex? I generally agree that the ciliary bands are likely homologous as you have them labeled in Figure 9 based on relative positions (and maybe type of cilia?) in the adults. But the reference to development to support this argument of homology needs to be explained in more detail.
Lines 305 – 308 Can you provide any suggestions about what this similarity might mean for their evolutionary origin, function?
Lines 311 – 313 You suggest that the three apical cells are the ASO in the dwarf males (which is lacking in the females), but then suggest that there are also sensory neurons in the cerebral ganglia of the dwarf males (e.g. here in the Discussion). Does this imply that some of the apical organ cells or some other sensory cells have become incorporated into the cerebral ganglia in the dwarf males? Is this the same for the females of the same species (i.e. do the females have sensory cells in their cerebral ganglia)? This also relates to my earlier question about what criteria were used to classify cells as sensory or not.
Lines 313 – 315 This is an example where a better explanation of the results in the Windoffer and Westheide paper would be helpful. How does finding six main cell in the cerebral ganglia of dwarf males contradict the previous findings? Were there more/fewer cells identified in the previous paper? Was the morphology different?
Lines 320 – 321 What is this assumption for homology with the nuchal organ based on? Morphology?
Lines 330 – 331 What is meant by “contacts with two sensory cells”? Is there evidence of a synapse between the sensory cells and the apical 5HT+ cell? If not, just having the cells be in close proximity is not really evidence of them communicating. This same comment applies to a few other places in the paper where proximity of neurons is used to imply a functional connection (e.g. lines 357 – 359).
Fig. 11 Please list what all of the abbreviations stand for. Alternatively, you could generate one abbreviations list for the whole paper to reduce redundancy.
A final diagram showing all of the elements of the nervous system described in this paper (e.g. both acetylated-tubulin and 5HT+ cells and neurites) together in one diagram from multiple views would be helpful.
Author Response
Point 1: Overview
The author describes morphology of the ciliary bands and nervous system in the dwarf male of the meiofaunal, paedomorphic species Dimorphilus gyrociliatus using anti-serotonin and anti-acetylated tubulin immunolabeling and confocal microscopy. Previous molecular works describing the adult nervous system and ciliary bands in D. gyrociliatus have largely focused on the females of this species. Overall, the data in this paper are an important contribution to our understanding of lophotrochozoan ciliary band evolution and the nervous system structure in an interstitial, paedomorphic annelid. However, as written, the Results section lacks sufficient explanation of the findings. Furthermore, comparisons with results for D. gyrociliatus dwarf males from other papers (e.g. Windoffer and Westheide 1988), females of the same species, and with other annelid species, both miniaturized and not, are needed to put this work in context and elucidate its significance and potential evolutionary implications.
Response 1: Thank you for carefully reading. I've dramatically changed the manuscript, includin introduction, results, discussion sections< added more details, schematic representations and comparisons.
Major comments
Point 2: The paper lacks detail in many areas (Intro, Results and Discussion), and the descriptions are fairly precursory. The figure panels are well-labeled with many different views, but having a clearer, written description of the data is necessary. I also think a better explanation of what was known about the morphology of the dwarf males prior to this study and how this work compares with other work (see above) is necessary.
Response 2: Thank you. I included detais on dwarf male morphology, based on Windoffer and Westheide papers and Kerbl paper.
Point 2: It is difficult for me to tell why all of the single, ciliated cells are considered sensory neurons. I do not see neurites or other morphology that would indicate a neuronal cell. Also, some explanation of the classification system being used would be helpful. For example, in lines 162 – 165, what makes these “type 1a cells”? These cells do not look different than the “type 1b cells” to me, so an explanation of how they were classified seems important. Is this based on specific morphological features?
Response 3: For this study I used two previous manuscripts Windoffer and Westheide, 1988a and b. They introduced the classification of sensory neurons, which includes 4 types: type 1a,b,c and type 2. They also created a detailed and helpful scheme, projection of the distribution of all these sensory neurons. that's why I consider these cells as sensory neurons. They actually have neurites, but it's easier to When I got confocal stacks, I analysed them and tried to find all described sensory neurons. Finally, I found them all.
Minor comments
Introduction
Point 4: lines 74 - 76 I think there is a verb missing in this sentence.
“In addition, during development, the prototroch nerve….”
Response 4: I've rewritten this phrase. The Introduction section was changed.
lPoint 5: ines 78 – 79 This sentence needs to be re-written. I am not certain what you mean by this statement. Are you saying that no ASO has been found within species belonging to Dinophiliformia? Or that no ASO has been visualized in females of D. gyrociliatus?
“D. gyrociliatus belongs to the lophotrochozoan Annelida, in whose representatives no ASO was mentioned.”
Response 5: Thank you. Actually I ment that in D. gyrociliatus no apical organ was mentioned. I deleted this sentence from the text, since Introduction was rewritten.
Point 6: Lines 80 – 81 Please add a few sentences about the morphology of the dwarf males. For example (mentioned below), it would be helpful to know if these dwarf males have a stomadeum and/or gut.
Response 6: Dwarf male lacks mouth, stomadeum and gut, according to present knowledge.
Methods
Point 7: I could not find the clone and dilution for the anti-5HT antibody. I also could not find the dilution used for phalloidin. How were the immunolabeled specimens stored before imaging? Were they stored in the 90% glycerol (in PBS?) at 10 oC before imaging?
Response 7: Thank you to mention about 5-HT antibodies: 1:2000 dilution , Immunostar, Hudson, WI, USA; 428002; rabbit; polyclonal; Product ID: 20080 . I included this information into method section and deleted information about phalloidin, since I never show it in figures. Specimens were stored in 90% glycerol in PBS at 10jC before imaging. It's mentioned in maethods.
Results
Point 8: In general, within the text, I would refer to the abbreviations for the structures along with the Figure panels. For example, the first time that you mention the “anterior ciliary field” line 147, I would refer to (avc, Figure 1 A–D). Also, I might provide a little more description of the ciliary fields. Later in the text you mention multi-ciliated cells that I think are part of the ciliary bands, but I wasn’t certain how this differed from the multi-ciliated cells that were labeled as “type 1c”.
Response 8: Thank you for advice, I followed this way and referred to the abbreviations for the structures as in your example. I also added a list of abbreviations at the end of the manuscript.
Point 9: line 154 I think you mean that the posterior ciliary field is located posterior to (rather than “below”) the anterior ciliary field. Also, from the images and diagrams, it looks like both the avc and the pvc are localized to the ventral face of the animal. Is this correct? If so, I would mention this in the text. I might also add “ventral” to the names for both.
Response 9: corrected, I added "ventral" to anterior and posterior ventral ciliary fields.
Point 10: line 161-162 Add a citation.
Response 10: I added a citation
Point 11: Keep the same nomenclature for the sensory neurons throughout. Sometimes they are referred to as “sensory neuron no. X” and then in the same sentence “sensory neuron # X”
Response 11: corrected, now I use sensory neuron №1-20.
Point 12: Line 234 and Figure 7 What is meant by “tubulin-like”? For the previous figures, cells were referred to as ‘labeled with anti-acetylated tubulin’. Usually, authors use the term “-like” for cross-reactive antibodies that may be labeling something other than their original antigen. But I do not see why the elements being described in this section are not just acetylated-tubulin positive. I just was not certain if this change in wording was meant to indicate some difference or not.
Response 12: corrected
Point 13: Lines 252 – 253 Why are these two pairs of 5HT+ cells considered “ganglia” and not neurons associated with the stomodeum as in other annelids? Do the dwarf males have a mouth and/or gut?
Response 13: Dwarf male has no moth or gut, according to present knowledge. So these neurons are not aasosiated with the stomodeum. But these neurons are similar with that of D. gyrociliatus females during the middle trochophore stage.
Point 14: Figure 1 It’s difficult to see panel “E”. I might make the white line around this inset a little thicker. For D, I am not certain what view is being shown (i.e. does “top view” refer to an “anterior view”). For the bottom panel of diagrams, I would add a horizontal line between these and the panels above so that it’s clear the “dorsal” and “top” labels associated with C and D, respectively, are separate from the diagrams. Otherwise, the diagrams and labels in this figure are very helpful.
Response14: Thank you, top view refers to anterior view. I added separation lines.
Point 15: Figures 2, 4 and 5 Why are the multi-ciliated cells referred to as “E10” and “E11”? This nomenclature is only used in the figure legends but is not described anywhere in the text that I can find.
Response 15: This nomenclature for multiciliated cells was introduces by Windoffer and Westheide. I used it in figure labeling, it helped me a lot to label sensory neurons. Since I've changed the manuscrip, I included this information in the Introduction section together with the schematic drawing.
Point 16: Figure 5 What is the cyan labeling in panel A?
Response 16: It is phalloidin staining. I deleted it from panel, since it was not the purpose of the figure.
Point 17: Figures 7 and 8 Add an explanation of what each color is showing (i.e. magenta is anti-5HT, red is…etc). Also, what does “co” refer to (copulatory organ)?
Response 17: thank you, I added color coding explanation: yellow-tubulin, magenta-5-HT, red-DAPI. Yes, co refers to copulatory organ. I added this information to figure legend and created a list of abbreviations.
Discussion
Point 18: Line 284 A better explanation of the results from the Windoffer and Westheide 1988 paper in the Discussion would be helpful. It is not very clear to me how your results compare to theirs.
Response 18: Thank you for the comment. I included more details to the Introduction and discussion sections
Point 19: Lines 290 – 291 I do not understand this statement as written. Why do the two sexes undergo ‘the same developmental program’? Why couldn’t there be initiation of a sex-specific developmental program that generated a non-homologous ciliary band in one but not the other sex? I generally agree that the ciliary bands are likely homologous as you have them labeled in Figure 9 based on relative positions (and maybe type of cilia?) in the adults. But the reference to development to support this argument of homology needs to be explained in more detail.
Response 19: I rewritten the Discussion section and deleted this sentence. There are no data on D. gyrociliatus dwarf male development, due to its extremely small size and high loss of samples during preparation. a possible homology between female and male ciliary structures is only my hypothesis. Of course it needs additional strong evidences.
Point 20: Lines 305 – 308 Can you provide any suggestions about what this similarity might mean for their evolutionary origin, function?
response 20: A homoplasy or homology
Poin 21: Lines 311 – 313 You suggest that the three apical cells are the ASO in the dwarf males (which is lacking in the females), but then suggest that there are also sensory neurons in the cerebral ganglia of the dwarf males (e.g. here in the Discussion). Does this imply that some of the apical organ cells or some other sensory cells have become incorporated into the cerebral ganglia in the dwarf males? Is this the same for the females of the same species (i.e. do the females have sensory cells in their cerebral ganglia)? This also relates to my earlier question about what criteria were used to classify cells as sensory or not.
Response 21: Sensory cells may have become incorporated into cerebral ganglia. The females do not have apical organ or sensory cells in their cerebral ganglia.
Point 22: Lines 313 – 315 This is an example where a better explanation of the results in the Windoffer and Westheide paper would be helpful. How does finding six main cell in the cerebral ganglia of dwarf males contradict the previous findings? Were there more/fewer cells identified in the previous paper? Was the morphology different?
Response22: I've rewritten the discussion. In my study fewer cells were identified, the morphology on the whole is the same.
Point 23: Lines 320 – 321 What is this assumption for homology with the nuchal organ based on? Morphology?
Response 23: I deleted this sentence. The assumption was based on morphology and position on the body.
Point 24: Lines 330 – 331 What is meant by “contacts with two sensory cells”? Is there evidence of a synapse between the sensory cells and the apical 5HT+ cell? If not, just having the cells be in close proximity is not really evidence of them communicating. This same comment applies to a few other places in the paper where proximity of neurons is used to imply a functional connection (e.g. lines 357 – 359).
Response24: I deleted this phrase , just mentioned he proximity of these cells.
Point 25: Fig. 11 Please list what all of the abbreviations stand for. Alternatively, you could generate one abbreviations list for the whole paper to reduce redundancy.
Response 25: corrected
Point 26: A final diagram showing all of the elements of the nervous system described in this paper (e.g. both acetylated-tubulin and 5HT+ cells and neurites) together in one diagram from multiple views would be helpful.
Response 27: I added this diagram into Discussion section.

Reviewer 3 Report
The manuscript by Elizaveta Fofanova is dedicated to the neuromorphology of the dwarf male of the enigmatic marine annelid Dimorphilus gyrociliatus. The results are in line with previous studies and add some new data to our understanding of the structure and evolution of dwarf males in dinophillids and male dwarfism in general. I would recommend the manuscript for publication, but only after a broad and substantial revision. Despite the fine and informative images and schemes the text looks like the author was in a hurry and had no time to check the manuscript before submitting.
I recommend the rewriting of the abstract and introduction sections. In the introduction one should at least mention the studies of D. gyrociliatus dwarf male nervous system morphology by Windoffer and Westheide (Windoffer R, Westheide W. The Nervous-System of the Male Dinophilus gyrociliatus (Annelida, Polychaeta) .1. Number, Types and Distribution Pattern of Sensory Cells. Acta Zool. 1988;69(1):55–64; Windoffer R, Westheide W. The nervous system of the male Dinophilus gyrociliatus (polychaeta, dinophilidae): II. Electron microscopical reconstruction of nervous anatomy and effector cells. J Comp Neurol. 1988;272:475–88). The previous information about general nervous system morphology and receptor cells should be provided to introduce the reader.
The results section should also be substantially rewritten. I recommend to add a short morphological description of the male. On the figures one can see much more morphological structures, not only ciliary fields, receptor cells and nervous system, so I ask the author to add this “gross morphological” information to make the article more friendly for the broad range of zoologists and evolutionary biologists who may not be familiar with dinophillid morphology.
Being the non-native English speaker I cannot judge about the language quality. Nevertheless, I would recommend a proofreading.
Please, find below a list of my comments to the manuscript text and figures.
Line 10: Here and below, why do you use “We”? You are the only author of this manuscript and may use singular.
Lines 25-27: “We used a standard immunochemical protocol in combination with anti-acetylated tubulin and anti-5 HT antibodies and confocal microscopy to visualize the ciliary structures and individual neurons.” Antibodies are usually the main actors in all standard immunochemical protocols, so in my opinion it is nonsense to write “immunochemical protocol in combination with antibodies”. It is usually the confocal microscopy that is combined with immunochemical labeling. The second point is about the “standard” protocol. Standard for what? I know various protocols with different permeabilisation, antigen unmasking tecniques, blocking and antibody incubation time as well as temperature regime. So I suggest rewrite the sentence e.g. “We used immunochemical detection of anti-acetylated tubulin and anti-5 HT antibodies in combination with confocal microscopy to visualize the ciliary structures and individual neurons” or something similar.
Line 28: While reading the abstract I usually hope to understand the main idea of the article and get to know the main results. The phrase “The nervous system of the male is complex” adds nothing to the description and in fact is the wasting of space. Moreover, the following sentence provides exhaustive information on the main features of NS organization.
Line 43: What is the current status of Dinophilids? If my memory doesn’t let me down, they were placed inside Sedentaria after Struck et al., 2015 (http://dx.doi.org/10.1016/j.cub.2015.06.007). If so, they should be mentioned in Sedentaria together with refs 1 and 2. Otherwise, dinophilids should be mentioned separately.
Lines 46-48. What unique features do you mention? Please name some of them and provide references. The features mentioned below (ciliary locomotion, protonephridia end so on) are not unique.
Line 51: Something wrong is the end of the sentence. You did not write a word about similarities with annelids here, so it is quite illogical to finish the sentence with “but unlike most Annelida do not have chaeta”.
Lines 62-64: This suggestion is only partially true. The nervous system may keep some conservative features, but it is at the same time a very functionally determined. See for example the article by Helm et al. (DOI: https://doi.org/10.1186/s12983-018-0280-y) about ladder-like appearance of the annelid nervous system. I do not argue with you, but please always keep it in mind.
Line 66: “…and nothing about their nervous system.” But what about two articles by Windoffer and Westheide (Windoffer R, Westheide W. The Nervous-System of the Male Dinophilus-Gyrociliatus (Annelida, Polychaeta) .1. Number, Types and Distribution Pattern of Sensory Cells. Acta Zool. 1988;69(1):55–64; Windoffer R, Westheide W. The nervous system of the male Dinophilus gyrociliatus (polychaeta, dinophilidae): II. Electron microscopical reconstruction of nervous anatomy and effector cells. J Comp Neurol. 1988;272:475–88). At least one of these works is cited in your article, so this sentence looks a bit weird.
Lines 74-77: A verb is missing.
Lines 78-79: I cannot agree with it. At least in Phyllodoce (DOI: 10.1002/cne.10488) Platynereis (DOI: 10.1186/s12983-017-0211-3), and Pomatoceros (DOI: 10.1186/1742-9994-3-16), and Malacoceros (DOI: 10.1186/s12862-020-01680-x) it was found.
Line 88: Seems like it should be “…immunoreactive system IN 3D and transverse sections…” or something like that.
Line 94: Urtica should be in italics.
Line 102: Once again: what is “standard”? I would suggest removal of this definition.
In Materials and methods section the description of anti-serotonin antibody and its dilution is missing.
Lines 147-161: First, I do not understand the division of the text in two paragraphs. Second, You use the receptors naming introduced by Windoffer and Westheide in “The Nervous-System of the Male Dinophilus-Gyrociliatus (Annelida, Polychaeta) 1. Number, Types and Distribution Pattern of Sensory Cells.” (Acta Zool. 1988;69(1):55–64), but the corresponding reference is missing. Third, if you refer to the nomenclature by Windoffer and Westheide, it would be much better to declare it in the very beginning of the results section, but definitely not in the end of the chapter.
Line 156: “The easiest to identify…” change to “The most prominent…”
Line 157: “neurons №9 (Figure 1B, C) on the dorsal side” change to “neurons №9 on the dorsal side (Figure 1B, C)”.
Line 162: When you describe different types of receptor cells, please provide at least short morphological descriptions for them (e.g. cell shape, number of cilia or any other information). I am familiar with the articles by Windoffer and Westheide, but other potential readers may not. This makes your data difficult to understand for non-specialists and even for specialists who did not deal with dinophilids.
Line 177: “These cells do not show a positive signal with 5- HT antibodies”. You have already mentioned this (Line 165).
Line 196: Please, use either “No” or “#” throughout the MS, but do not mix them.
Line 206: I suggest changing “multiple ciliated” to “multiciliated”.
Line 221: change 2-type to type 2 or type II to better fit the nomenclature of Windoffer and Westheide.
Line 233: Pleae, check figure numbering. Should it the reference to Figure 7 here? In my opinion It should be figure 6.
Line 235: I compared the text with Figure 6A and 6A’ and cannot call these cells “lateral”. Maybe “paramedial” or something like that. According to your illustrations true lateral cells do not possess such processes.
Line 275: You write about a single process of the unpaired cell, but I cannot find it on your figures. As far as I can see, there are two processes that come from the single stem (Fig 7A’, C’).
Lines 276-278: Please match specific images to the text, not the whole figures. Each figure has several images and finding the corresponding structures is time consuming, and, to be honest, boring. In addition, on Fig 8F should it be sn14 instead of sn4?
Lines 282-284: I did not understand the sentence. Perhaps you mean “We provided the first detailed data on the organization of 5- HT immunoreactive nervous system of D. gyrociliatus dwarf male and clarified the landmark of ciliary cells and sensory neurons”?
Line 284 as well as 289: Once again, there are two articles by Windoffer and Westheide. The text of the results section clearly indicates that you know both these papers, so cite both of them.
Line 289: “structures of cilia” sounds a bit strange. Maybe “ciliary structures”, or even “ciliary fields or pathces” would better fit your idea.
Line 290: “Since males and females undergo the same developmental program…” If so, they would possess similar morphology, but that is not the case. I recommend you to rewrite this sentence.
Line 293: Why do you split the paragraph in two parts? They both share the same idea and should be united. But perhaps this is an artefact of the automatic layout.
Line 307: D. gyrociliatus should be in italics
Line 308: should be “[31].” But I would also know your opinion whether this similarity has any evolutionary sense (homology/homoplasy).
Line 311: Do the cerebral ganglia in your manuscript correspond the frontal ganglia from Windoffer and Westheide? If so, please explain why do you introduce this new term?
Line 316: What are “these sensory cells”?
Line 343: I really doubt that ASO (apical sensory organ) contains 5HT and FMRFamide-like cells only. This looks like Texas sharpshooter fallacy, just because everybody checked only these substances. The whole sentence looks like a citation. If so, please provide a reference.
Lines 368-370: Confocal microscopy is now a routine method for nervous and muscular system reconstructions. 3D and sharp optical sections were key features of the confocal microscopy from the very beginning. What is the advances here? The specimen preparation is easier and faster comparing with what? I suggest removal of the whole sentence.
Figures:
I have two major points addressed to the figure legends. First, check the abbreviations. Not all the abbreviations in the illustrations are deciphered in the corresponding legends. Alternatively, you may add a separate abbreviations list in the end of the manuscript.
Second, the figure legends are to “scarce”, they add almost no information about the figures. Please make them a bit more detailed.
Figure 1 The bottom line of schemes is not labeled. In further illustrations the schemes are labeled by letters with strokes.
Figure 1 B, C – 9 should be sn9
Figure 1 B, the scale bar is missing
Asterisks labeling nephridia on Figure 1 B-D are too small, please enlarge to make them more visible
Figure 2
Why the scheme on A have two dissection lines? The same question is addressed also for Figure 6 A
The dotted rectangles on A and B have different letter sizes (c and D). Please make the font sizes more uniform.
Figure 3
What is the scale bar sizes on D-J?
See my comment above on the asterisks size.
Figure 4
The abbreviation “go” is not deciphered in the figure legend.
Figure 5
Some of the abbreviations intersect with bright areas of the images which makes them difficult to read and even hard to find! e.g. cells E10 and E11 on A. Some scale bars are missing.
Figure 6:
Perhaps you could sign dorsal and ventral commissures on B separately (e. g. dc and vc), to facilitate understanding of the figures.
The abbreviations “dn”, “gl”, and “vb” are not deciphered in the figure legend.
I do not understand the meaning of the dotted box on C labeled “Fig. 7”.
Figure 7
See my comment above on the asterisks size.
Please check the abbreviations. I did not find “co”, “spd”, and “vb”.
The double arrows on A’-D’ are too small and look like single. Please enlarge them or use other pictograms, e.g. arrowheads.
Figure 8
The scale bar on D’ is missing.
Please check the abbreviations and keep in mind my comment above on the asterisks size.
Figure 9
Please check the abbreviations
Author Response
Reviewer3
The manuscript by Elizaveta Fofanova is dedicated to the neuromorphology of the dwarf male of the enigmatic marine annelid Dimorphilus gyrociliatus. The results are in line with previous studies and add some new data to our understanding of the structure and evolution of dwarf males in dinophillids and male dwarfism in general. I would recommend the manuscript for publication, but only after a broad and substantial revision. Despite the fine and informative images and schemes the text looks like the author was in a hurry and had no time to check the manuscript before submitting.
Point 1: I recommend the rewriting of the abstract and introduction sections. In the introduction one should at least mention the studies of D. gyrociliatus dwarf male nervous system morphology by Windoffer and Westheide (Windoffer R, Westheide W. The Nervous-System of the Male Dinophilus gyrociliatus (Annelida, Polychaeta) .1. Number, Types and Distribution Pattern of Sensory Cells. Acta Zool. 1988;69(1):55–64; Windoffer R, Westheide W. The nervous system of the male Dinophilus gyrociliatus (polychaeta, dinophilidae): II. Electron microscopical reconstruction of nervous anatomy and effector cells. J Comp Neurol. 1988;272:475–88). The previous information about general nervous system morphology and receptor cells should be provided to introduce the reader.
Response 1; I've rewritted the abstract, Introduction and discussion sections. I mentioned precious studies by Windoffer and Westheide and used their classification of sensory neurons.
Point 2: The results section should also be substantially rewritten. I recommend to add a short morphological description of the male. On the figures one can see much more morphological structures, not only ciliary fields, receptor cells and nervous system, so I ask the author to add this “gross morphological” information to make the article more friendly for the broad range of zoologists and evolutionary biologists who may not be familiar with dinophillid morphology.
Response 2: I put more information to the Introduction and Results sections.
Point 3: Being the non-native English speaker I cannot judge about the language quality. Nevertheless, I would recommend a proofreading.
Response 3: The manuscript was proofread before the submitting. Now I reorginised the manuscript and proofread it again.
Please, find below a list of my comments to the manuscript text and figures.
Point 4: Line 10: Here and below, why do you use “We”? You are the only author of this manuscript and may use singular.
Response: I got used to "we" and I consulted with colleagues, moreover I mention them in acknowledgements section.
Point 5: Lines 25-27: “We used a standard immunochemical protocol in combination with anti-acetylated tubulin and anti-5 HT antibodies and confocal microscopy to visualize the ciliary structures and individual neurons.” Antibodies are usually the main actors in all standard immunochemical protocols, so in my opinion it is nonsense to write “immunochemical protocol in combination with antibodies”. It is usually the confocal microscopy that is combined with immunochemical labeling. The second point is about the “standard” protocol. Standard for what? I know various protocols with different permeabilisation, antigen unmasking tecniques, blocking and antibody incubation time as well as temperature regime. So I suggest rewrite the sentence e.g. “We used immunochemical detection of anti-acetylated tubulin and anti-5 HT antibodies in combination with confocal microscopy to visualize the ciliary structures and individual neurons” or something similar.
response 5: I'we rewritten and reorginised Method section.
Point 6: Line 28: While reading the abstract I usually hope to understand the main idea of the article and get to know the main results. The phrase “The nervous system of the male is complex” adds nothing to the description and in fact is the wasting of space. Moreover, the following sentence provides exhaustive information on the main features of NS organization.
Response 6: I've rewritten the abstract section.
Point 7: Line 43: What is the current status of Dinophilids? If my memory doesn’t let me down, they were placed inside Sedentaria after Struck et al., 2015 (http://dx.doi.org/10.1016/j.cub.2015.06.007). If so, they should be mentioned in Sedentaria together with refs 1 and 2. Otherwise, dinophilids should be mentioned separately.
Response 7: Dinophilids belong to a newly emerged Dinophiliformia group which is a sister group to Pleistoannelida : Martín-Durán, J.M.; Vellutini, B.C.; Marlétaz, F.; Cetrangolo, V.; Cvetesic, N.; Thiel, D.; Henriet, S.; Grau-Bové, X.; Carrillo-Baltodano, A.M.; Gu, W.; et al. Conservative route to genome compaction in a miniature annelid. Nat Ecol Evol 2021, 5, 231–242, doi:10.1038/s41559-020-01327-6.
Worsaae, K.; Kerbl, A.; Domenico, M.D.; Gonzalez, B.C.; Bekkouche, N.; Martínez, A. Interstitial Annelida. Diversity 2021, 13, 77, doi:10.3390/d13020077.
POint 8: Lines 46-48. What unique features do you mention? Please name some of them and provide references. The features mentioned below (ciliary locomotion, protonephridia end so on) are not unique.
Response 8; rewritten the Introduction and deleted this phrase.
Point 9: Line 51: Something wrong is the end of the sentence. You did not write a word about similarities with annelids here, so it is quite illogical to finish the sentence with “but unlike most Annelida do not have chaeta”.
Response 9: I deleted this phrase and changed the Introduction section .
Poinyt 10: Lines 62-64: This suggestion is only partially true. The nervous system may keep some conservative features, but it is at the same time a very functionally determined. See for example the article by Helm et al. (DOI ) about ladder-like appearance of the annelid nervous system. I do not argue with you, but please always keep it in mind.
Response 10: I read this article carefully< cited it and designed a schematic drawing based on this article with the addition of Dinophiliformia. And mentioned the difference between males and females nervous systems.
Point 11: Line 66: “…and nothing about their nervous system.” But what about two articles by Windoffer and Westheide (Windoffer R, Westheide W. The Nervous-System of the Male Dinophilus-Gyrociliatus (Annelida, Polychaeta) .1. Number, Types and Distribution Pattern of Sensory Cells. Acta Zool. 1988;69(1):55–64; Windoffer R, Westheide W. The nervous system of the male Dinophilus gyrociliatus (polychaeta, dinophilidae): II. Electron microscopical reconstruction of nervous anatomy and effector cells. J Comp Neurol. 1988;272:475–88). At least one of these works is cited in your article, so this sentence looks a bit weird.
Response 11: These two articles helped me to define and label sensory and ciliary cells. I cited both. I ' ve changed the manuscript drastically.
Point 12: Lines 74-77: A verb is missing.
Respone: I've rewritten the introduction section.
Point 13: Lines 78-79: I cannot agree with it. At least in Phyllodoce (DOI: 10.1002/cne.10488) Platynereis (DOI: 10.1186/s12983-017-0211-3), and Pomatoceros (DOI: 10.1186/1742-9994-3-16), and Malacoceros (DOI: 10.1186/s12862-020-01680-x) it was found.
Response 13: That's actually what I meant. Apical organ is present in Phyllodoce, Platynereis, Pomatoceros< Malacoceros and Owenia. But no apical organ signs in Dimorphilus gyrociliatus were found. And in present aricle one of results a possible apical organ made up of three cells in D. gyrociliatus male.
Point 14: Line 88: Seems like it should be “…immunoreactive system IN 3D and transverse sections…” or something like that.
Response 14: Corrected , I've rewritten "present data on the distribution of cilia, sensory neurons, and the 5-HT-like immunoreactive system in 3D reconstructions and cross-sections and compare our data with those of D. gyrociliatus females and previous results on metazoan dwarf males"
Point 15: Line 94: Urtica should be in italics.
Response 15: corrected
Point 16: Line 102: Once again: what is “standard”? I would suggest removal of this definition.
Response 16: Thank you. I removed this phrase.
Point 17: In Materials and methods section the description of anti-serotonin antibody and its dilution is missing.
Response 17: Corrected. I added the description of anti-serotonin antibody and dilution Immunostar, Hudson, WI, USA; 428002; rabbit; polyclonal; Product ID: 20080, diluted 1:2000.
Point 18: Lines 147-161: First, I do not understand the division of the text in two paragraphs. Second, You use the receptors naming introduced by Windoffer and Westheide in “The Nervous-System of the Male Dinophilus-Gyrociliatus (Annelida, Polychaeta) 1. Number, Types and Distribution Pattern of Sensory Cells.” (Acta Zool. 1988;69(1):55–64), but the corresponding reference is missing. Third, if you refer to the nomenclature by Windoffer and Westheide, it would be much better to declare it in the very beginning of the results section, but definitely not in the end of the chapter.
Response 18: Thank you. I've rewritten the Results section and put the receptors naming introduced by Windoffer and Westheide with the referece in the very beginning of the results. "The neural elements are described in the following order: type 1a, type 1b, type 1c, and type 2 receptors; tubulin-positive nervous system; and 5-HT-like immunoreactive nervous system. All sensory neurons and structures are described using the classification invented by Windoffer and Westheide [3,4]. Individual neurons were recognized by determining their position according to the schematic diagram from Westheide and Windoffer."
Point 19: Line 156: “The easiest to identify…” change to “The most prominent…”
Response 19: corrected
Point 20: Line 157: “neurons №9 (Figure 1B, C) on the dorsal side” change to “neurons №9 on the dorsal side (Figure 1B, C)”.
Response 20: Corrected.
Point 21: Line 162: When you describe different types of receptor cells, please provide at least short morphological descriptions for them (e.g. cell shape, number of cilia or any other information). I am familiar with the articles by Windoffer and Westheide, but other potential readers may not. This makes your data difficult to understand for non-specialists and even for specialists who did not deal with dinophilids.
Response 21: I' ve wertitten the manuscript and short morphological description is present in Introduction section. The authors also gave an accurate and detailed description of the nervous system, which includes about 68 neuronal cell bodies. Most of these neurons were classified as sensory based on their position and morphology. Sensory neurons were numbered and categorized into four types according to their morphology: type 1a (monociliated with collar), type 1b (monociliated without collar), type 1c (multiciliated without collar), and type 2 receptor cells (with non-emergent cilia).
Point 22: Line 177: “These cells do not show a positive signal with 5- HT antibodies”. You have already mentioned this (Line 165).
Response 22: Corrected
Point 23: Line 196: Please, use either “No” or “#” throughout the MS, but do not mix them.
Response 23: corrected I changed into sensory neuron №...
Point 24: Line 206: I suggest changing “multiple ciliated” to “multiciliated”.
Response 24: yes, it is. Thank you
Point 25: Line 221: change 2-type to type 2 or type II to better fit the nomenclature of Windoffer and Westheide.
Response 25: corrected
Point 26: Line 233: Pleae, check figure numbering. Should it the reference to Figure 7 here? In my opinion It should be figure 6.
Response26: thank you. I added new figures and checked numbering.
Point 27: Line 235: I compared the text with Figure 6A and 6A’ and cannot call these cells “lateral”. Maybe “paramedial” or something like that. According to your illustrations true lateral cells do not possess such processes.
Response 27: Corrected
Point 28: Line 275: You write about a single process of the unpaired cell, but I cannot find it on your figures. As far as I can see, there are two processes that come from the single stem (Fig 7A’, C’).
Response 28: corrected
Point 29: Lines 276-278: Please match specific images to the text, not the whole figures. Each figure has several images and finding the corresponding structures is time consuming, and, to be honest, boring. In addition, on Fig 8F should it be sn14 instead of sn4?
Response29: on Fig 8F sn4, sn 14 is in the posterior part of the body, close to sn 15.
Point 30: Lines 282-284: I did not understand the sentence. Perhaps you mean “We provided the first detailed data on the organization of 5- HT immunoreactive nervous system of D. gyrociliatus dwarf male and clarified the landmark of ciliary cells and sensory neurons”?
Response 31: Yes. that's actually what i meant.
Point 31: Line 284 as well as 289: Once again, there are two articles by Windoffer and Westheide. The text of the results section clearly indicates that you know both these papers, so cite both of them.
Response 31: Corrected
Point 32: Line 289: “structures of cilia” sounds a bit strange. Maybe “ciliary structures”, or even “ciliary fields or pathces” would better fit your idea.
Response 32: I changed into "ciliary structures"
Point 33:Line 290: “Since males and females undergo the same developmental program…” If so, they would possess similar morphology, but that is not the case. I recommend you to rewrite this sentence.
Response 33: I deleted this sentence.
Point 34: Line 293: Why do you split the paragraph in two parts? They both share the same idea and should be united. But perhaps this is an artifact of the automatic layout.
Response 34: I rewritten the discussion. Perhaps this was an artifact of the layout
Point 35: Line 307: D. gyrociliatus should be in italics
Response 35: corrected
Point 36:Line 308: should be “[31].” But I would also know your opinion whether this similarity has any evolutionary sense (homology/homoplasy).
Response 37: it may be a homoplasy
Point 37: Line 311: Do the cerebral ganglia in your manuscript correspond the frontal ganglia from Windoffer and Westheide? If so, please explain why do you introduce this new term?
Response 37: I corrected the term. Yes ganglia in my manuscript correspond with the frontal ganglia from Windoffer and Westheide.
Point 38: Line 316: What are “these sensory cells”?
Response38: I deleted this phrase and rewritte discussion
Point 39: Line 343: I really doubt that ASO (apical sensory organ) contains 5HT and FMRFamide-like cells only. This looks like Texas sharpshooter fallacy, just because everybody checked only these substances. The whole sentence looks like a citation. If so, please provide a reference.
Response 39: I deleted these sentences and rewritten the paragraph on apical organ. Buy in most studied annelids aso contains 5-HT and FMRFamide-like cells and other cells with unknown transmitter.
Point 40: Lines 368-370: Confocal microscopy is now a routine method for nervous and muscular system reconstructions. 3D and sharp optical sections were key features of the confocal microscopy from the very beginning. What is the advances here? The specimen preparation is easier and faster comparing with what? I suggest removal of the whole sentence.
Response 40: Thank you< I removed the whole sentence.
Figures:
Point 41: I have two major points addressed to the figure legends. First, check the abbreviations. Not all the abbreviations in the illustrations are deciphered in the corresponding legends. Alternatively, you may add a separate abbreviations list in the end of the manuscript.
Response 41: Thank you. I checked abbreviations and added a list of abbreviations at the end of the manuscript.
Point 42: Second, the figure legends are to “scarce”, they add almost no information about the figures. Please make them a bit more detailed.
Response 42: I gave more details .
Point 43: Figure 1 The bottom line of schemes is not labeled. In further illustrations the schemes are labeled by letters with strokes.
Response 43: I reorganised figures and labeled schemes .
Point 44:Figure 1 B, C – 9 should be sn9
Response 44: corrected
Point 45: Figure 1 B, the scale bar is missing
Response 45: corrected
Point 46: Asterisks labeling nephridia on Figure 1 B-D are too small, please enlarge to make them more visible
Response 46: I made a bigger asterisk and verified it on other figures
Figure 2
Point 47: Why the scheme on A have two dissection lines? The same question is addressed also for Figure 6 A
Response 47: Corrected, I deleted extra dissection line markers.
Point 48: The dotted rectangles on A and B have different letter sizes (c and D). Please make the font sizes more uniform.
Response 48: corrected
Figure 3
Point 49:What is the scale bar sizes on D-J?
Response 49: scale bar on D-J size is 1.6 microns, the letters are quite big< so I wrote in the right part of the figure.
Point 50: See my comment above on the asterisks size.
Response 50: corrected
Figure 4
Point 51: The abbreviation “go” is not deciphered in the figure legend.
Response 51: go stands for genital opening. I included it to the legend and to the list of abbreviations.
Figure 5
Point 52: Some of the abbreviations intersect with bright areas of the images which makes them difficult to read and even hard to find! e.g. cells E10 and E11 on A. Some scale bars are missing.
Response 52: I reorganized all the figures, now cells E10 and E11 much easier to read. Scale bars on close ups are the same 1.5 microns, so I put bars and at the end gave the size 1.5 micron.
Figure 6:
Point 53: Perhaps you could sign dorsal and ventral commissures on B separately (e. g. dc and vc), to facilitate understanding of the figures.
Response 53: corrected
Point 54: The abbreviations “dn”, “gl”, and “vb” are not deciphered in the figure legend.
Response 54: Corrected. These abbreviations are now deciphered and included into the list of abbreviations.
Point 55: I do not understand the meaning of the dotted box on C labeled “Fig. 7”.
Response 55: I meant that the same cells are shown on Fig 7, they are 5-HT+ cells.
Figure 7
Point 56: See my comment above on the asterisks size.
Response 56: Corrected
Point 57: Please check the abbreviations. I did not find “co”, “spd”, and “vb”.
Response 57: Thank you< Ichecked abbreviations and included them into list of abbreviations
Point 58: The double arrows on A’-D’ are too small and look like single. Please enlarge them or use other pictograms, e.g. arrowheads.
Response 58: thank you< I changed the double arrows into arrowheads
Figure 8
Point 59: The scale bar on D’ is missing.
Response 59: Corrected
Point 60: Please check the abbreviations and keep in mind my comment above on the asterisks size.
Response 60: corrected
Figure 9
Point 61: Please check the abbreviations
Response 61: corrected

Reviewer 4 Report
This article describes the nervous system architecture and ciliary patterns in a small enigmatic annelid Dinomorphilus gyrociliatus where data on its morphology is sparse. This work has evolutionary implications in understanding how animal nervous systems may have evolved and crucial for understanding conserved nervous system architectures within Annelida. This is a comprehensive amount of work where the author(s) have described morphology of the cilia, distribution of peripheral sensory neurons and the serotonin immunoreactive neurons in this poorly studied annelid. Such characterization of nervous system and ciliary architectures have been recently performed on other annelids in an attempt to understand evolutionary patterns of neural development in annelids which can inform the understanding of the overall evolution of nervous systems. This manuscript adds to that information that in combination with similar studies would be instrumental in understanding how nervous systems may have evolved. Similar studies from several poorly studied groups in combination are required in future. The manuscript is well written, and figures are clear and easy to follow. I therefore recommend this manuscript to be accepted in the journal “Biology” with some minor revisions. My suggested revisions are enlisted below:
General comments:
a. This work has quite a few evolutionary implications – it would make the manuscript stronger if the author can emphasize the evolutionary implications of this work by comparing against other annelids such as Owenia, Capitella teleta and Platynereis dumeriliii. There are some general questions that can be addressed for better understanding the significance of this work - How is Dinomorphilus gyrociliatus phylogenetically related to other annelids? Which other annelid species are they most closely related to? Please highlight how similar or different the nervous system architecture is with other closely related annelids.
b. Also, if possible, please have a section and potentially a figure or diagram describing the development of Dinomorphilus gyrociliatus and when do nervous system development begin. That would help orient the readers about what developmental stages they are looking at here in the manuscript. It would be useful to learn how far in development are these neurons already present.
c. As Dinomorphilus gyrociliatus is a non-model organism, not everyone would be familiar with its anatomy. It would be greatly useful if the author can add a diagram and figure at the beginning of the manuscript with the anatomy of this enigmatic annelid. That would help orient the readers with what is anterior, posterior, dorsal and ventral.
Specific comments:
(1) Introduction:
Line 61: It would be really useful if you can elaborate how Dinomorphilus gyrociliatus is related to the ancestral group as of the current hypothesis. Please cite a manuscript that describes the phylogenetic relationship of Dinomorphilus gyrociliatus with other annelids.
Line 78: This sentence may be redundant with previous sentences above as the author has already convinced that Dinomorphilus gyrociliatus belongs to Annelida. Maybe incorporate this sentence before.
(2) Materials and methods: Please have subheadings for your individual paragraphs. There are several steps that were done – it would be easier to follow if they are divided up into subheadings
(3) Results:
i. Line 157, 158 – The sensory neurons are numbered in the figure as sn9, sn15 etc. which is easy to follow. However, in the text, the sensory neurons are referred to as “No_9”, “No-15” etc. This may be an artefact of the version that was available to me as a reviewer, but in case it’s not, please use consistent nomenclature. If it’s labeled as “sn1”, “sn2” etc. in the figure, please use the same in the text as well. Please keep this consistent throughout the manuscript.
ii. A little more elaboration on the significance of the various experiments conducted would make the manuscript stronger. For example, what was the rationale behind analyzing 5’HT immunoreactivity. For each section, please mention the rationale behind the experiments performed.
iii. Did you observe differences in morphologies between mono-ciliated and multi-ciliated cells?
iv. In Figure 1, the bottom panel is not labeled as “E”. Please label that panel.
v. In Figure 1, bottom panel, move the word “ventral” more to the right
vi. Line 160 – is a citation needed here?
vii. Section 3.2, Line 162 – It’s not clear if the definition of Type 1a, Type 1b, and Type 1c cells are classified in this study or was this previously defined in other publications. If this study defines these subtypes, please elaborate how these categories were defined, i.e. based on what criteria. If the categories were previously defined, please provide appropriate citations.
viii. Please label the Type1a, Type1b, Type1c, and 2-Type cells in the respective figures. It would be better to see where these cells are located in the animal in the figures themselves.
ix. Figures have been well-labelled and looks clean and easy to interpret. However, labels and designations need some work. In all figures, please label which color indicates which immunoreactivity.
x. For figures 2-7, it would make more sense to reorder the figure panels or change the designation of the figure panels. For example, A’ should come after A, B’ after B and so on. Right now, A’, B’, C’ succeeds some other letter. Please label the figures in order.
xi. In all figures, please make sure all abbreviations are spelled out or elaborated. For example, in Figure 5, the abbreviation “test” or “c” is not spelled out.
xii. For Figure 8, it needs a Panel A. Figure 8 starts from B, please change to panel A or add a new panel A.
xiii. For the confocal micrographs, it may be useful to show individual channels in black and white with appropriate labels of the antibody used in these experiments.
xiv. In lines 237, 240 and other, it would be useful to add arrows or arrowheads to your figures and refer to them in text in order to guide the viewers which specific cell types you are talking about.
(4) Discussion:
i. Please highlight evolutionary implications based on comparisons on other previously studied annelids. How do their ciliary and 5’HT architecture compare to that of Dinomorphilus gyrociliatus. You may consider comparing against individual annelids and speculate about an ancestral annelid state.
ii. Add a couple of sentences on how this study is important for shaping future projects using Dinomorphilus gyrociliatus.
Author Response
This article describes the nervous system architecture and ciliary patterns in a small enigmatic annelid Dinomorphilus gyrociliatus where data on its morphology is sparse. This work has evolutionary implications in understanding how animal nervous systems may have evolved and crucial for understanding conserved nervous system architectures within Annelida. This is a comprehensive amount of work where the author(s) have described morphology of the cilia, distribution of peripheral sensory neurons and the serotonin immunoreactive neurons in this poorly studied annelid. Such characterization of nervous system and ciliary architectures have been recently performed on other annelids in an attempt to understand evolutionary patterns of neural development in annelids which can inform the understanding of the overall evolution of nervous systems. This manuscript adds to that information that in combination with similar studies would be instrumental in understanding how nervous systems may have evolved. Similar studies from several poorly studied groups in combination are required in future. The manuscript is well written, and figures are clear and easy to follow. I therefore recommend this manuscript to be accepted in the journal “Biology” with some minor revisions. My suggested revisions are enlisted below:
General comments:
Point 1: a. This work has quite a few evolutionary implications – it would make the manuscript stronger if the author can emphasize the evolutionary implications of this work by comparing against other annelids such as Owenia, Capitella teleta and Platynereis dumeriliii. There are some general questions that can be addressed for better understanding the significance of this work - How is Dinomorphilus gyrociliatus phylogenetically related to other annelids? Which other annelid species are they most closely related to? Please highlight how similar or different the nervous system architecture is with other closely related annelids.
Response 1: The species Dimorphilus gyrociliatus belongs to the newly emerged group Dinophiliformia. This group is a sister group to Pleistoannelida.
I put this information at the beginning of an Introduction section.
Point 2: b. Also, if possible, please have a section and potentially a figure or diagram describing the development of Dinomorphilus gyrociliatus and when do nervous system development begin. That would help orient the readers about what developmental stages they are looking at here in the manuscript. It would be useful to learn how far in development are these neurons already present.
Response 2: Thank you for your interest to D. gyrociliatus. Actually quite little is known about their development. The data on cleavage and gastrulation are present in Nelsons' manuscript (). The data on neurogenesis in two dinophilid species D. vorticoides (both sexes) and D. gyrociliatus (females only) are present in my last year manuscript at PeerJ. We demonstrated that the first nerve cells appear during the early trochophore strage in both species. D. gyrociliatus neurogenesis is special for its' the early anterior FMRFa+ cell. So in terms of development we defined three stages: the early, middle and late trochophore stages. I included the schematic drawing into discussion section and compared D. gyrociliatus female nervous system organisation with that of a male
Point 3: c. As Dinomorphilus gyrociliatus is a non-model organism, not everyone would be familiar with its anatomy. It would be greatly useful if the author can add a diagram and figure at the beginning of the manuscript with the anatomy of this enigmatic annelid. That would help orient the readers with what is anterior, posterior, dorsal and ventral.
Response 3: Thank you. I added some information on the anatomy. Thus, a schematic drawing of multiciliated cells from ciliary fields is added to the Introduction section. Additional pagargaph added to the results section. I put figure illustrating the position of testis, seminal vesicles, and details on glandomuscular copulatory organ with its' stylet gland cells.
Specific comments:
(1) Introduction:
Point 4: Line 61: It would be really useful if you can elaborate how Dinomorphilus gyrociliatus is related to the ancestral group as of the current hypothesis. Please cite a manuscript that describes the phylogenetic relationship of Dinomorphilus gyrociliatus with other annelids.
Response 4; I've rewritten and reorganised the manuscript text completely. At the very beginning I put a scheme of annelid phylogenetic tree with newly emerged group Dinophiliformia and cited relevant literature:
Martín-Durán, J.M.; Vellutini, B.C.; Marlétaz, F.; Cetrangolo, V.; Cvetesic, N.; Thiel, D.; Henriet, S.; Grau-Bové, X.; Carrillo-Baltodano, A.M.; Gu, W.; et al. Conservative route to genome compaction in a miniature annelid. Nat Ecol Evol 2021, 5, 231–242, doi:10.1038/s41559-020-01327-6.
Worsaae, K.; Kerbl, A.; Domenico, M.D.; Gonzalez, B.C.; Bekkouche, N.; Martínez, A. Interstitial Annelida. Diversity 2021, 13, 77, doi:10.3390/d13020077.
Point 5: Line 78: This sentence may be redundant with previous sentences above as the author has already convinced that Dinomorphilus gyrociliatus belongs to Annelida. Maybe incorporate this sentence before.
Respoinse 5: Corrected
Point 6: Materials and methods: Please have subheadings for your individual paragraphs. There are several steps that were done – it would be easier to follow if they are divided up into subheadings
Response 6: Corrected< I put three subheadings: Animal handling, Fixation and immunostaining and Microscopy and Image processing.
(3) Results:
Point 7: i. Line 157, 158 – The sensory neurons are numbered in the figure as sn9, sn15 etc. which is easy to follow. However, in the text, the sensory neurons are referred to as “No_9”, “No-15” etc. This may be an artefact of the version that was available to me as a reviewer, but in case it’s not, please use consistent nomenclature. If it’s labeled as “sn1”, “sn2” etc. in the figure, please use the same in the text as well. Please keep this consistent throughout the manuscript.
Response 7: Thank you for comment. I've rewrtitten the results section and in the text refer sensory neuron № 1(sn1, FigureX). And I was consistent throughout the manuscript.
Point 8: ii. A little more elaboration on the significance of the various experiments conducted would make the manuscript stronger. For example, what was the rationale behind analyzing 5’HT immunoreactivity. For each section, please mention the rationale behind the experiments performed.
Response:8I rewritten the Introduction section and explained the significance of analyzing 5-HT immunoreactivity.
Point 10: iii. Did you observe differences in morphologies between mono-ciliated and multi-ciliated cells?
Response 10: Yes, there is a difference between mono- and multiciliated cells. Describing sensory neurons distribution I used Windoffer and Westheide classification and nomenclature. Sensory neurons №12 and №13 have 3 cilia, that corresponds with Windoffer and Westheide results.
Other multiciliated cells belong to locomotory ciliary fields. Each of these cells possess
about 150 cilia per cell< and these cilia are located in strong order. The same cilia are in locomotory ciliary structures of females.
Point 11: iv. In Figure 1, the bottom panel is not labeled as “E”. Please label that panel.
Response 11: I reorginised the figures and labeled all thepanels.
Poin 12: v. In Figure 1, bottom panel, move the word “ventral” more to the right
Response 12: Corrected
Point 13: vi. Line 160 – is a citation needed here?
Response 13: I changed discussio section dramatically< and added citations.
Pint 14: vii. Section 3.2, Line 162 – It’s not clear if the definition of Type 1a, Type 1b, and Type 1c cells are classified in this study or was this previously defined in other publications. If this study defines these subtypes, please elaborate how these categories were defined, i.e. based on what criteria. If the categories were previously defined, please provide appropriate citations.
Response 14: These categories were introduced in Windoffer and Wesheide publication, based on morphology. I add this information to the Introduction section.
: Windoffer, R.; Westheide, W. The nervous system of the male Dinophilus gyrociliatus (Annelida: Polychaeta). I. Number, types and distribution pattern of sensory cells. Acta Zoologica 1988, 69, 55–64, doi:10.1111/j.1463-6395.1988.tb00901.x.
Point 15: viii. Please label the Type1a, Type1b, Type1c, and 2-Type cells in the respective figures. It would be better to see where these cells are located in the animal in the figures themselves.
Response 15: Actually figures orginised in that way. Each figure represents a certain receptor type as well as parts of the Results section
Point 16: ix. Figures have been well-labelled and looks clean and easy to interpret. However, labels and designations need some work. In all figures, please label which color indicates which immunoreactivity.
Response 16: Corrected
Point 17: x. For figures 2-7, it would make more sense to reorder the figure panels or change the designation of the figure panels. For example, A’ should come after A, B’ after B and so on. Right now, A’, B’, C’ succeeds some other letter. Please label the figures in order.
Response 17: I've reordered panels in most of the figures.
Point 18: xi. In all figures, please make sure all abbreviations are spelled out or elaborated. For example, in Figure 5, the abbreviation “test” or “c” is not spelled out.
Response 18: corrected
Point 19: xii. For Figure 8, it needs a Panel A. Figure 8 starts from B, please change to panel A or add a new panel A.
Response 19: I labelled the wholemount as A, because I want reader to look at this panel firstly, and then other panels in order according to labelling.
Point 20: xiii. For the confocal micrographs, it may be useful to show individual channels in black and white with appropriate labels of the antibody used in these experiments.
Response 21: For ciliary distribution figures it is like that, only tubulin channel is present. But in other Figures I gave tubulin +DAPI staining in most cases. I think reader could download the figure and use PS to study individual channels. Or I could submit raw data on figshare as individual channels.
Point 22: xiv. In lines 237, 240 and other, it would be useful to add arrows or arrowheads to your figures and refer to them in text in order to guide the viewers which specific cell types you are talking about.
Response 22: For Neurons related to CNS I changed double arrows into arrowheads. And refer to them.
(4) Discussion:
Point 23: i. Please highlight evolutionary implications based on comparisons on other previously studied annelids. How do their ciliary and 5’HT architecture compare to that of Dinomorphilus gyrociliatus. You may consider comparing against individual annelids and speculate about an ancestral annelid state.
Response 23: I changed the discussion section almost completely and highlighted evolutionary implications based on comparisons < I also put a scheme based on Helm, C.; Beckers, P.; Bartolomaeus, T.; Drukewitz, S.H.; Kourtesis, I.; Weigert, A.; Purschke, G.; Worsaae, K.; Struck, T.H.; Bleidorn, C. Convergent evolution of the ladder-like ventral nerve cord in Annelida. Front Zool 2018, 15, 36, doi:10.1186/s12983-018-0280-y.
Point 24: ii. Add a couple of sentences on how this study is important for shaping future projects using Dinomorphilus gyrociliatus.
Response 24: I wrote that nothing is know about D. gyrociliatus male development.

Round 2
Reviewer 3 Report
The author addressed all my comments and corrections, and I am satisfied with them. The author made a great work improving the manuscript text as well as figures. The text looks more clear and understandable for a broad range of specialists.The figures were also improved and well-structured. Nevertheless I still have some minor comments to the revision.
First of all, please double check the text of the manuscript since it has some typos, such as missing spaces and non-correspondances between figures and their references in the text. I Found some mistakes (e.g. lines 458 or 428) but not sure that this is all.
Line 91: Anterior ventral ciliary field
Line 112: You write "penis ganglia", but later call them "penile ganglia" (line 117). I think you should choose one of the spellings and follow it throughout the text.
Line 167: ...male dwarf fish of D. gyrociliatus. What does "fish" mean?
Line 408: should it be "type 2 cells"?
Line 428: ...neurons №9 (double arrows, Figure 7... Maybe Figure 10 and And these should be "arrowhead".
Line 449: check the correspondance between B and B' ("c" in B and "dc, vc" in B')
Line 458: Should the reference be 10A or 10D?
Line 574: What figure do you mean?
Lines 607-632: Too many separate paragraphs in my opinion. You may easily unite them without damaging the story.
Lines 692-693: I cannot agree with such formulating. Acetylated alpha-tubulin is a structural protein, thus almost all nerve cells should a priori be tubulin-positive. The point is that these cells did not react with the antibodies tested in that study, isn't it? You should reformulate the sentence.
Author Response
The author addressed all my comments and corrections, and I am satisfied with them. The author made a great work improving the manuscript text as well as figures. The text looks more clear and understandable for a broad range of specialists. The figures were also improved and well-structured. Nevertheless I still have some minor comments to the revision.
Point 1: First of all, please double check the text of the manuscript since it has some typos, such as missing spaces and non-correspondances between figures and their references in the text. I Found some mistakes (e.g. lines 458 or 428) but not sure that this is all.
Response 1: Corrected. I double checked the text of the manuscript and fixed typos, missing spaces and extra spaces. Actually it may have been occurred during different versions of Office. I used Yandex to male track changes file. And like the first round I could open the latest version with Google office only.
I hope, now everything is fine. I checked MS with various versions of office.
Point 2: Line 91: Anterior ventral ciliary field
Resonse 2: Corrected.
Point 3: Line 112: You write "penis ganglia", but later call them "penile ganglia" (line 117). I think you should choose one of the spellings and follow it throughout the text.
Response 3: Corrected, I've chosen penis ganglia.
Point 4: Line 167: ...male dwarf fish of D. gyrociliatus. What does "fish" mean?
Response 4:It is a typo. I deleted it. Thank you!
Point 5: Line 408: should it be "type 2 cells"?
Response 5: Corrected. Thank you!
Point 6: Line 428: ...neurons №9 (double arrows, Figure 7... Maybe Figure 10 and And these should be "arrowhead".
Response 6:Corrected, I changed into arrowhead and added reference to Figure 10).
Point 7: Line 449: check the correspondance between B and B' ("c" in B and "dc, vc" in B')
Response 7: Corrected, I added dc and vc to B.
POint8: Line 458: Should the reference be 10A or 10D?
Response 8: Both references are correct. If my line numbering is correct. I checked the Figure 10 and corrected pvc into avc, so now both references refer to paired 5-HT+ neurons on the ventral side
Point 9: Line 574: What figure do you mean?
Response 9: I meant Figures 9 and 12. I added these references to the text.
